

# Variations in export production, lithogenic sediment transport and iron fertilization in the Pacific sector of the Drake Passage over the past 400 ka

María H. Toyos[1,2,3], Gisela Winckler [4,5], Helge W. Arz[6], Lester Lembke-Jene[3], Carina B. Lange[2,7,8],
Gerhard Kuhn[3], and Frank Lamy[3]

[1] Programa de Postgrados en Oceanografía, Departamento de Oceanografía, Facultad de Ciencias Naturales y Oceanográficas, Universidad de Concepción, Concepción, Chile
[2] Centro de Investigación Dinámica de Ecosistemas Marinos de Altas Latitudes (IDEAL), Universidad Austral de Chile, Valdivia, Chile
[3] Alfred-Wegener-Institut, Helmholtz-Zentrum für Polar und Meeresforschung, Bremerhaven, Germany
[4] Lamont-Doherty Earth Observatory, Columbia University, Palisades, NY 10964
[5] Department of Earth and Environmental Sciences, Columbia University, New York, NY 10027
[6] Leibniz-Institut für Ostseeforschung Warnemünde (IOW), Rostock-Warnemünde, Germany
[7] Departamento de Oceanografía and Centro Oceanográfico COPAS Sur-Austral, Universidad de Concepción, Chile
[8] Scripps Institution of Oceanography, La Jolla, California 92037

*Correspondence to*: Maria H Toyos (mtoyos@udec.cl)

**Abstract.** Changes in Southern Ocean export production have broad biogeochemical and climatic implications. Specifically, iron fertilization likely increased subantarctic nutrient utilization and enhanced the efficiency of the biological pump during
glacials. However, past export production in the subantarctic Southeast Pacific is poorly documented, and its connection to Fe fertilization, potentially related to Patagonian Ice Sheet dynamics is unknown. We report on biological productivity changes over the past 400 ka, based on a combination of $^{230}Th_{xs}$-normalized and stratigraphy-based mass accumulation rates of biogenic barium, organic carbon, biogenic opal, and calcium carbonate as indicators of paleo-export production in a sediment core upstream of the Drake Passage. In addition, we use fluxes of iron and lithogenic material as proxies for terrigenous matter, and
thus potential micronutrient supply. Stratigraphy-based mass accumulation rates are strongly influenced by bottom-current dynamics, which result in variable sediment focussing or winnowing at our site. Carbonate is virtually absent in the core, except during peak interglacial intervals of the Holocene, and Marine Isotope Stages (MIS) 5 and 11, likely caused by transient decreases in carbonate dissolution. All other proxies suggest that export production increased during most glacial periods, coinciding with high iron fluxes. Such augmented glacial iron fluxes at the core site were most likely derived from glaciogenic
input from the Patagonian Ice Sheet promoting the growth of phytoplankton. Additionally, glacial export production peaks are also consistent with northward shifts of the Subantarctic and Polar Fronts, which positioned our site south of the Subantarctic Front and closer to silicic acid-rich waters of the Polar Frontal Zone, as well as a with a decrease in the diatom utilization of Si relative to nitrate under Fe-replete conditions. However, glacial export production near the Drake Passage was lower than in the Atlantic and Indian sectors of the Southern Ocean, which may relate to complete consumption of silicic acid in the study



area. Our results underline the importance of micro-nutrient fertilization through lateral terrigenous input from South America rather than aeolian transport, and exemplify the role of frontal shifts and nutrient limitation for past productivity changes in the Pacific entrance to the Drake Passage.

## 1 Introduction

The Southern Ocean (SO) plays an essential role in modulating glacial-interglacial variations of atmospheric $p$CO$_2$ (Sigman

et al., 2010). Increased biological export production, fueled by enhanced iron (Fe) fertilization in a more stratified glacial SO, is thought to have been a key driver of increased deep marine carbon storage (Jaccard et al., 2013). In the SO, the wind-driven, eastward flowing Antarctic Circumpolar Current (ACC) enhances the air-sea exchange of CO$_2$ and the upwelling of nutrient- and CO$_2$-rich subsurface water masses (e.g., Marshall and Speer, 2012). Its flow is concentrated along several fronts, which are the Subantarctic Front (SAF), Polar Front (PF), and Southern ACC Front (SACCF; Orsi et al., 1995). These fronts act as

barriers, inhibiting the exchange of the upwelled waters and their associated nutrients with neighboring fronts, and therefore also represent the limits between geochemical provinces (Chapman et al., 2020; Paparazzo, 2016). Furthermore, these SO fronts are not stationary and their positions have been shown to change on seasonal to orbital timescales (e.g. Gille, 2014; Kemp et al., 2010). The major constriction for the ACC flow and SO fronts is the Drake Passage (DP) located between the southern tip of South America and the Antarctic Peninsula. At the DP entrance, a consistent pattern of glacial reduction of the

ACC throughflow has been previously linked to a northward shift of the SAF (Lamy et al., 2015; Toyos et al., 2020). Today, the SO represents the major region in the world ocean where the efficiency of the biological carbon pump is low. Incomplete nutrient utilization arises from a combination of high overturning rates, yielding fast replenishment of most macronutrients to the photic zone, and an Fe limitation, restricting phytoplankton growth (Boyd et al., 2012; Moore et al., 2013). Therefore, mean chlorophyll concentrations above 2 mg m$^{-3}$ are only found within 50 km off a continental or island

coastline (Fig 1, Graham et al., 2015). Specifically, in the Subantarctic Zone (SAZ; the area between the SAF and the Subtropical Front), waters feature excess nitrate relative to silicate (Dugdale et al., 1995) and Fe limits phytoplankton growth (Boyd et al., 1999). Thus, diatom production is relatively low, coccolithophores control primary production (Rigual-Hernández et al., 2015), and the biological pump is predominantly driven by carbonate producing organisms (Honjo, 2004). In contrast, the region between the SAF and the PF, known as the Polar Frontal Zone (the PFZ), is characterized by high abundances of

large diatoms (Kopczynska et al., 2001), and by relatively low mean chlorophyll concentrations (Fig 1, Graham et al., 2015). The modern low export production in the SAZ contrasts with increased biological activity during glacials, predominantly fueled by enhanced Fe fertilization (e.g., Anderson et al., 2014; Jaccard et al., 2013; Kohfeld et al., 2005; Lamy et al., 2014; Martínez-Garcia et al., 2009; Thöle et al., 2019). However, other mechanisms such as a more efficient diatom growth, shifts in the dominant plankton types, and increased nutrient utilization due to the lack of Fe limitation may also explain the glacial

increase in export production, which may lead to an increase in the biological carbon pump's efficiency in the SAZ (François et al., 1997; Galbraith and Skinner, 2020; Matsumoto et al., 2014). Of the proposed mechanisms, only Fe fertilization





contributed significantly to the lowering of atmospheric $CO_2$, explaining the last 30 to 50 ppm of atmospheric $CO_2$ drawdown during the last glacial period (Kohfeld et al. 2005, Martin, 1990). It is widely believed that the Fe driving this fertilization of the glacial subantarctic SO is primarily delivered via aeolian dust (e.g., Martínez-Garcia et al., 2009, 2014). This idea has been

corroborated by reconstructions of past glacial-interglacial variability in the southern westerly wind belt, which indicate a glacial strengthening and an equatorward migration/extension (Ho et al., 2012; Kohfeld et al., 2013; Lamy et al., 2014).

In addition to dust input, especially in the vicinity of continents, Fe may also be brought to the surface ocean via continental runoff, iceberg transport or meltwater, and/or coastal upwelling (De Baar and De Jong, 2001). An increasing number of studies have recently shown that icebergs could provide a comparable and/or larger amount of bioavailable Fe than dust to the SO

(e.g., Hopwood et al., 2019), and therefore significantly influence primary productivity (Wu and Hou, 2017). For instance, Fe associated with subglacial meltwater and icebergs stimulates and enhances marine primary productivity in ecosystems around the Antarctic Ice Sheet (Laufkötter et al., 2018; Vernet et al., 2011), and in the SAZ during the Last Glacial Maximum (LGM), where a combination of high ice discharge and slower iceberg melting due to colder sea surface temperatures increased the supply of Fe-rich terrigenous material (Wadham et al., 2019). It is well known that during glacial stages, erosion increased

significantly, enhancing physical weathering and, in turn, the sediment supply to the deep ocean. However, despite comprising a non-trivial portion of the SO, past export production in the southeastern Pacific, and its connection to direct Fe fertilization via meltwater and/or icebergs, potentially related to Patagonian Ice sheet (PIS) dynamics, remains unexplored. The subantarctic Pacific in the vicinity of the entrance to the DP is proximal to Patagonia, but likely does not receive substantial dust from this region, given the prevailing wind direction of the southern westerlies. In fact, Patagonian dust is predominantly transported

eastward to the SO's Atlantic sector (Li et al., 2008, 2010). Therefore, the DP entrance location might provide a unique opportunity to explore the PIS's potential as a direct source of Fe for fertilization of the southeastern Pacific during glacial intervals.

In this study, we reconstruct and characterize export production changes off Patagonia in the subantarctic SE Pacific over the past 400 ka and investigate its link to Fe fertilization and SO frontal shifts. We use a combination of $^{230}Th_{xs}$-normalized and

stratigraphy-based mass accumulation rates of the lithogenic fraction, Fe, excess barium, i.e. the fraction of Ba that is not supplied by terrigenous material ($Ba_{xs}$), total organic carbon (TOC), biogenic opal, and carbonate ($CaCO_3$) from sediment core PS97/093-2 located near the SAF at the DP entrance (Fig 1). While $Ba_{xs}$ and TOC may reflect the integrated total export production, $CaCO_3$ and biogenic opal indicate changes in the export production related to specific organisms. These are mainly coccolithophores and foraminifera for carbonate and diatoms and radiolarians for opal (Dymond, 1992; Paytan, 2009). We

show that variations in export production were closely linked to terrigenous sediment and Fe delivery from the Patagonian hinterland, glacier dynamics and fundamental climate/ocean changes in the region. Lastly, we discuss potential underlying mechanisms, and evaluate our results with respect to published SAZ paleoproductivity reconstructions from other SO sectors.



## 2 Study area

Our site PS97/093 is located in the subantarctic Southeast Pacific, at the western entrance to the DP (57° 29.94' S; 70° 16.48' W). The DP, located between South America and the Antarctic Peninsula, is 850 km wide and represents the major geographical constriction for ACC transport into the Atlantic (Gordon et al., 1978). At present, the geostrophic transport of the ACC in the DP is associated with the SAF and PF, where strong surface and bottom velocities have been observed (Meredith et al., 2011; Renault et al., 2011). Furthermore, glacial-interglacial variability in bottom current circulation of *ca.*

16% during the last 1.3 Ma. has been reported at our coring site (Toyos et al., 2020). At present, Site PS97/093 is bathed in Circumpolar Deep Water (CDW), which consists of a mixture of aged North Atlantic Deep Water that enters the SO through the Atlantic, recirculated Pacific and Indian Deep waters, and dense bottom waters from Antarctica (e.g., Carter et al., 2008). The carbonate ion concentration of CDW is nearly constant at $85\pm5$ μmol/kg (Broecker and Clark, 2001), and it is slightly undersaturated with respect to calcium (Key et al., 2004). In the DP, the seafloor features, the position of the core southern

westerly wind belt, and nutrient concentrations define the location of the ACC fronts (e.g., Ferrari et al., 2014; Meredith et al., 2011; Paparazzo et al., 2016). Here, the PF is associated with the northern expression of the silicate front, indicating the geographical boundary between the silicate-poor to the north of it and silicate-rich waters to the south (Freeman et al., 2018, 2019).Our sediment core was retrieved *ca*. 40 km NW of the present-day position of the SAF, and *ca.* 350 km NW of the PF, within the main flow of the ACC (Fig 1b).

The DP is the region of the SO with the largest vertical velocities associated with topographically induced upwelling (Graham et al., 2015). Nevertheless, the DP is not considered an area of intensive phytoplankton growth, as illustrated by low chlorophyll concentrations (Fig 1; Demidov et al., 2011). Primary production is characterized by phytoplankton blooms in austral spring (Demidov et al., 2011), when Fe availability is the critical factor for the occurrence of the blooms (De Baar et al., 1995). Additionally, a lack of dissolved silica compared to phosphorous may limit the growth of siliceous phytoplankton species north

of the PF (mainly diatoms, Demidov et al., 2011; Freeman et al., 2019). An analysis of the composition of the surface sediments shows that in the DP a gradual change occurs from low organic carbon and high carbonate content in the SAZ to opal-rich sediments at the Polar Front zone and in the Permanently Open Ocean Zone (Cárdenas et al., 2019).

In the northern part of the DP, the provenance of terrigenous materials is restricted to proximal source regions, specifically southern Patagonia (Wu et al., 2019). Furthermore, analyses of surface sediments from the eastern South Pacific and DP have

excluded a westward dust transport pathway from southern South America dust sources to this region (Wengler et al., 2019), as well as a substantial modern dust contribution from either Patagonia or Australia to the DP (Wu et al., 2019). After the Great Patagonian Glaciation (*ca.* 1 Ma), in at least five major glaciations, the Patagonian Andes were covered by a continuous mountain ice sheet, extending from 37°S to the Cape Horn, and to the Pacific Patagonian Shelf, with most ice likely calving into the Pacific Ocean on the western side, south of Chiloe island (Fig. 1; Davies et al., 2020; Gowan et al., 2021; Rabassa,

2008; Rabassa et al., 2011). As a result, at the southernmost Chilean continental margin, higher ice-rafted debris (IRD) contents





occurred during cold intervals, interpreted as advances of the PIS (Caniupán et al., 2011). The distance between core PS97/093-2 and the PIS during the last glacial (that reached ~56ºS, Glasser et al., 2008) was *ca.* 180 km (Fig 1b).

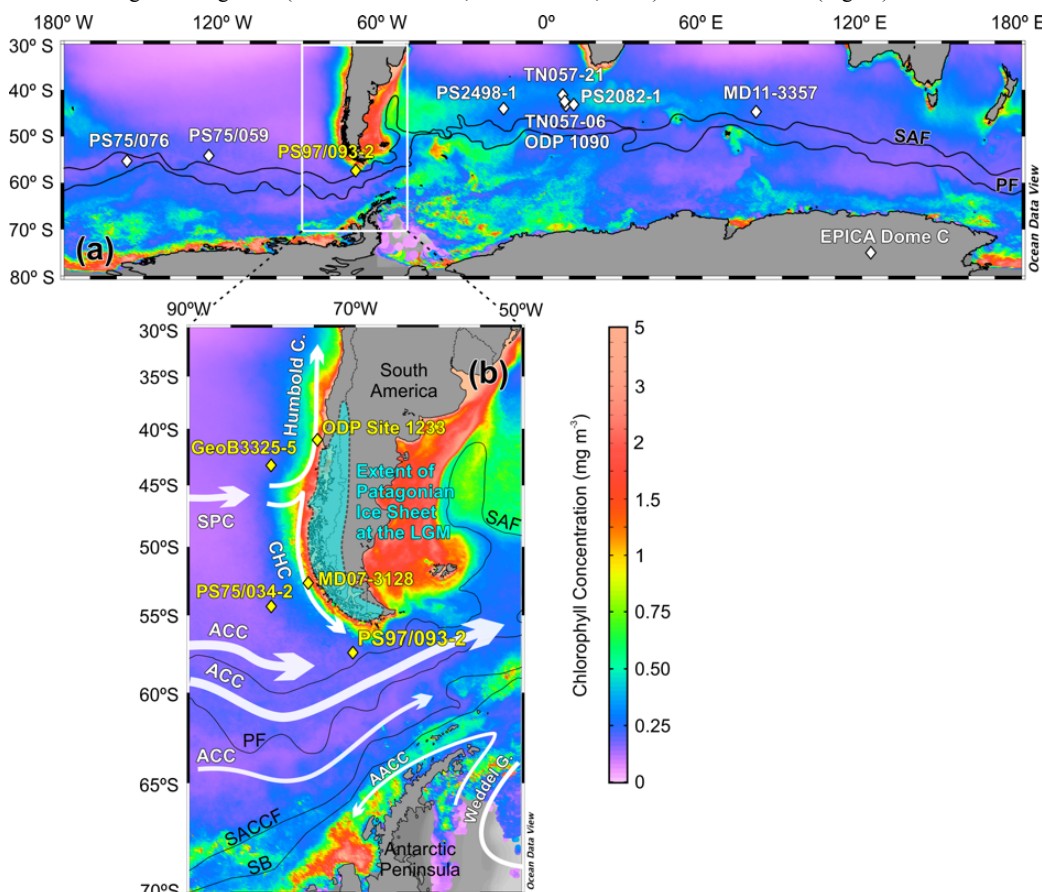

**Figure 1: (a) Map of the Southern Ocean mean chlorophyll-a concentrations for the years 2009–2019 with core locations. Yellow**
**diamond indicates the location of core PS97/093-2 (this study), and white diamonds the location of published records discussed in the text: PS75/076 and PS75/059 (Lamy et al. 2014); PS2498-1, TN057-21, and TN057-06 (Anderson et al., 2014); ODP1090 (Martínez-García et al. 2014); PS2082-1 (Frank et al. 1999; Nürnberg, 1997); MD11-3357 (Thöle et al., 2019), and EPICA Dome C ice Core (Lambert et al., 2008). (b) Map of the Drake Passage mean chlorophyll-a concentrations for the years 2009–2019 showing the location of PS97/093-2 (this study), MD07-3128 (Caniupán et al., 2011), PS75/034-2 (Ho et al., 2012) , GeoB3325-5 (Tapia et al.,**
**2021), and ODP Site 1233 (Lamy et al., 2010; yellow diamonds), cyan area indicates the extension of the Patagonian Ice Sheet at the Last Glacial Maximum based on Glasser et al. (2008), white arrows show trajectories of the Antarctic Circumpolar Current (ACC), South Pacific Current (SPC), Cape Horn Current (CHC), Humboldt Current (Strub et al., 2019), and Antarctic Coastal Current (AACC, Deacon, 1984). Black lines mark ACC modern fronts (Orsi et al., 1995). SAF, Subantarctic Front; PF, Polar Front; SACCF, southern Antarctic Circumpolar Current Front, and SB, southern boundary of the ACC. We used the MODIS-Aqua Level-3**
**Mapped Chlorophyll Data Version 2018 (data/10.5067/AQUA/MODIS/L3M/CHL/2018), in 4 km resolution monthly mean chlorophyll-a concentrations between March 2009 and March 2019 (available from NASA Ocean Color website, https://oceancolor.gsfc.nasa.gov/l3/) and Ocean Data View for visualization (Schlitzer 2021) (Schlitzer, Reiner, Ocean Data View, odv.awi.de, 2021).**



## 3 Material and methods

Piston core PS97/093-2 was retrieved from the Pacific entrance of the Drake Passage (57º 29.94' S; 70º 16.48' W; 3781 m
water depth; 16.45 m length; Fig. 1) during expedition PS97 "Paleo Drake" with R/V Polarstern (Lamy, 2016). Lithologically,
different types of hemipelagic sediments occur at this location that vary in composition on orbital timescales (Lamy, 2016).
Sediments from the interglacials MIS 11, MIS 5 and the Holocene are primarily composed of calcareous oozes (nannofossil
or foraminifera-nannofossil oozes) with minor concentrations of diatoms. In contrast, glacials and sediments from MIS 9, 7
and 3 are clayey silt with rare biogenic components, intercalated with some layers of diatomaceous fine-grained clayey silts.

### 3.1 Age model


The age model for core PS97/093-2 was taken from Toyos et al. (2020) and consists of a two-step approach: 1) establishment
of a preliminary age model based on biostratigraphic time markers from calcareous nannofossils and diatoms, and 2) fine-
tuning of the high-resolution XRF-derived elemental Fe and Ca counts and $CaCO_3$ contents to the LR04 benthic $\delta^{18}O$ stack
(Lisiecki and Raymo, 2005), using the AnalysSeries software (Paillard et al., 1996). For tuning, we assumed that low Fe
contents characterize interglacial periods, whereas high contents represent glacials. Additionally, XRF Ca counts and $CaCO_3$
percentages were used for additional tuning in the intervals where they are present.

### 3.2 Bulk sediment parameters and geochemistry

Total carbon and nitrogen (TC, TN) were quantified using a CNS analyzer (Elementar Varia EL III) at the Alfred-Wegener-
Institute, Bremerhaven (AWI) using 100 mg of freeze-dried and homogenized sediments. Total organic carbon (TOC) contents
were measured with a carbon-sulfur determinator (CS-2000, ELTRA) after the removal of inorganic carbon (total inorganic
carbon) by adding 37% (vol/vol) of hydrochloric acid. $CaCO_3$ was calculated employing the standard equation Eq. (1):

$$CaCO_3 \text{ [wt.\%]} = (TC \text{ [wt.\%]} - TOC \text{ [wt.\%]}) * 8.333 \qquad (1)$$

Biogenic opal was determined at the Laboratory of Paleoceanography, University of Concepción (UdeC), Chile. The alkaline
extraction was conducted following the procedure described by Mortlock and Froelich, (1989), but using NaOH as a digestion
solution (Müller and Schneider, 1993). Between fifty to seventy milligrams of freeze-dried sediment were first treated with
10% $H_2O_2$ and 1N HCl, and then extracted with 1M NaOH (40 mL; pH~13) at 85 °C for five hours. The analysis was carried
out by molybdate-blue spectrophotometry. Values are expressed as biogenic opal by multiplying the Si (%) by 2.4 (Mortlock
and Froelich, 1989). We did not correct for the release of extractable Si from coexisting clay minerals, and thus biogenic opal
values could be overestimated (Schlüter and Rickert, 1998). Biogenic opal was also measured at AWI Bremerhaven using the
sequential leaching method of Müller and Schneider (1993) at much lower temporal resolution, and offsets between the
overlapping data sets were observed. For terrigenous contents exceeding 75%, opal concentrations measured at UdeC are
consistently 3–5% higher than those measured at AWI. When the lithogenic content was below 40%, the inter-lab difference



was less than 1%. Despite the difference in values, both records show a similar pattern of variability. Given the importance of high-resolution data, we here use the opal results from UdeC. Dry bulk densities were quantified on a total of 162 samples

with a gas pycnometer (Micromeritics AccPyc II 1340) at AWI Bremerhaven, using the density measurements of freeze-dried and homogenized bulk sediment samples and calculated by incorporating the water content of the samples.

The archive half of core PS97/093-2 was measured with an AVAATECH X-Ray Fluorescence Core Scanner (XRF-CS) at AWI Bremerhaven for high-resolution semi-quantitative element intensities of Ca, Fe, Ba and Ti at 0.5 cm resolution (0.5 x 1.2 cm measurement area, slit size down- and across-core). Three consecutive runs were performed with tube voltages of

10 kV (no filter), 30 kV (Pd-thick filter) and 50 kV (Cu filter), a current of 0.15, 0.175 and 1 mA, acquisition times of 10 s, 15 s and 20 s, respectively. Raw data were processed using Canberra *Eurisys'* iterative least squares software (WIN AXIL) package. To obtain a high-resolution $CaCO_3$ record we calibrated the XRF Ca intensities with the bulk sediment $CaCO_3$ measurements ($r^2$=0.92, n=157, P<0.0001) (Fig S1).

### 3.3 Elements and U/Th isotope analysis

Concentrations of Fe, Ti, and Ba, along with U/Th isotopes, were determined at Lamont-Doherty Earth Observatory (LDEO). Freeze dried samples (100 mg) were spiked with a $^{236}U - ^{229}Th$ solution, followed by complete acid digestion (Fleisher and Anderson, 2003). Digests were taken up in 10 ml of 0.5 M $HNO_3$, and subsequently split in two aliquots. First, 0.4 ml was diluted again in 0.5 M $HNO_3$ to get a final dilution of 2000x that was used for the determination of Al, Fe, Ti and Ba concentrations. For the determination of U/Th isotopes, the remainder of the initial dilution (~ 9.6 ml) was utilized, with U/Th

purification achieved via Fe-coprecipitation and anion exchange chromatography following the methodology of Fleisher and Anderson, (2003). To check for reproducibility and for quality control purposes, an internal sediment standard (VOICE Internal MegaStandard (VIMS)) was run in each batch. U/Th isotopes were measured on a Thermo Scientific Element 2 ICP-MS, and absolute elemental concentrations of Fe, Ti, and Ba were determined using an ICP-OES. Finally, to obtain a high-resolution Fe record, we calibrated the 3-point smoothed Fe XRF intensities with bulk sediment (wt%) Fe from our discrete samples

($r^2$=0.90, n=132, P<0.0001) (Fig. S1).

### 3.4 Excess barium

$Ba_{xs}$ was calculated as:

$$Ba_{xs}= Ba_{total} - (Ti_{total}*[Ba/Ti]_{detrital}) \qquad \text{Eq. (2):} \qquad\qquad\qquad (2)$$

where $Ba_{total}$ is the total measured Ba, $Ti_{total}$ is the total measured Ti, and $[Ba/Ti]_{detrital}$ is the ratio of Ba and Ti in crustal material

(assumed here to be 0.126 after Turekian and Wedepohl, 1961). This methodology assumes that 1) the major source of elemental Ba to deep-sea sediments is marine barite, and 2) terrigenous material has a known and constant Ba/Ti ratio (e.g.,



Winckler et al., 2016). Our $Ba_{xs}$ record was obtained calibrating the 3-point smoothed Ba and Ti XRF intensities with bulk sediment (wt%) Ba ($r^2$=0.76, n=132, P<0.0001) and Ti ($r^2$=0.79, n=132, P<0.0001) (Fig. S1).

### 3.5 Mass accumulation rates

Mass accumulation rates of individual sediment components (Fe, $Ba_{xs}$, TOC, $CaCO_3$ and biogenic opal) were calculated by using the $^{230}Th_{xs}$ normalization method (Bacon, 1984; Francois et al., 2004). $^{230}Th$ is produced in the water column by decay of $^{234}U$, and has a short residence time, settling quickly to the underlying sediments by proximal scavenging. As such, the flux of $^{230}Th$ scavenged from the water column is considered to be nearly equal to its production rate. Therefore, the $^{230}Th_{xs}$-normalized mass accumulation rate (MAR) for a given sample can be determined by Eq (3):

$$MAR = \frac{\beta \times z}{\left[ ^{230}Th_{xs}^0 \right]},$$    (3)

where $\beta \times z$ is the integrated $^{230}Th$ production in the overlying water column that depends on the water depth (z), and $^{230}Th_{xs}^0$ is the measured $^{230}Th$ activity after corrections for i) $^{230}Th$ supported by $^{238}U$ in detrital sediments, ii) $^{230}Th$ supported by authigenic $^{238}U$ from the seawater, and iii) radioactive decay of $^{230}Th$ since deposition. $^{230}Th_{xs}$- normalized mass accumulation rates of individual components were obtained by multiplying their respective concentrations by the MAR. Here, we use the

MAR back to *ca.* 400 ka, although the errors increase in older sediments due to uncertainties in the lithogenic and authigenic corrections.

Additionally, stratigraphy-based bulk mass accumulation rates (BMAR) of individual components were obtained by multiplying the concentration of the respective component by the linear sedimentation rate and the dry bulk density.

### 3.6 Calculation of focusing factor

The degree of sediment focusing ($\Psi$) was calculated following the approach of Suman and Bacon, (1989) Eq (4):

$$\Psi = \left( \int_{r1}^{r2} {}^{230}Th_{xs}^0 \, \rho_r dr \right) / \beta_z (t_2 - t_1) ,$$    (4)

where $\rho_r$ is the dry bulk density (g/cm$^3$), $^{230}Th_{xs}^0$ is the concentration of excess $^{230}Th$ in the sediment corrected for decay since deposition, $t_1$ and $t_2$ are the corresponding ages (kyr) of sediment depths $r_1$ and $r_2$ (cm), and $\beta_z$ (dpm/cm$^2$/kyr) is the integrated $^{230}Th$ production in the overlying water column from $^{234}U$ decay. $\Psi > 1$ indicates sediment focusing, whereas $\Psi < 1$ denotes

sediment winnowing. Values of $\Psi = 1$ indicate that the amount of $^{230}Th$ buried in the sediment is equivalent to the amount of $^{230}Th$ produced in the water column.



### 3.7 Lithogenic content

The concentrations of lithogenic material were calculated using $^{232}$Th (e.g., Anderson et al., 2014; Winckler et al., 2008) assuming that i) $^{232}$Th is exclusively of detrital origin (Francois et al., 2004), and ii) the $^{232}$Th concentration of terrigenous
material is relatively constant . $^{232}$Th-derived lithogenic MAR were calculated by dividing the average $^{232}$Th concentration of lithogenic material from Patagonia (9 µg/g, McGee et al., 2016).

Additionally, the lithogenic content was also determined by subtraction of the biogenic sediment components from the total bulk, Eq (5):

$$lithogenic\ (wt\%) = 100 - [biogenic\ opal\ (wt\%) + CaCO_3\ (wt\%) + (2TOC\ (wt\%))], \tag{5}$$

**4 Results**

Carbonate content is almost zero throughout most of the record, peaking only during MIS 11 (~80%, which coincides with a prominent layer of nannofossil ooze, Lamy, 2016), MIS 5 (~40%), and the Holocene (~40%, Fig 2). In contrast, percentages of lithogenic material ($^{232}$Th-derived, and those obtained by subtracting biogenic sediment components from the total bulk) and Fe (wt%) show minima during these interglacials, and higher values during the rest of the record. Overall, the $^{232}$Th-based
lithogenic fraction is 10% lower than the lithogenic fraction obtained by subtraction (Fig 2b). The $Ba_{xs}$ content shows glacial-interglacial variability with higher values during interglacials, whereas the percentage of TOC does not display glacial-interglacial variability, and values increase gradually from MIS 5e to the Holocene. The biogenic opal percentage shows strong similarities with the lithogenic and Fe records with pronounced minima of less than four percent during MIS11, MIS 5, and the Holocene, and relatively high values during the rest of the record (~ 7.5–10 %).
$CaCO_3$ and $Ba_{xs}$ BMAR records vary in parallel with percentages of carbonate and $Ba_{xs}$ respectively, in contrast with all other BMAR of individual components that show no strong similarities with the corresponding percentages (Fig 2). From MIS 11 to MIS 6/5 transition all other BMAR records generally co-vary, showing two small peaks during MIS 9 and MIS 7, and a larger peak during Termination II that continues through the beginning of MIS 5. However, from MIS 4 to the Holocene there are some differences between all records; the lithogenic, Fe and opal BMAR records increase gradually from MIS 4 to MIS 2,
and decrease during the Holocene, whereas TOC BMAR increases gradually from MIS 4 to the Holocene (Fig 2).



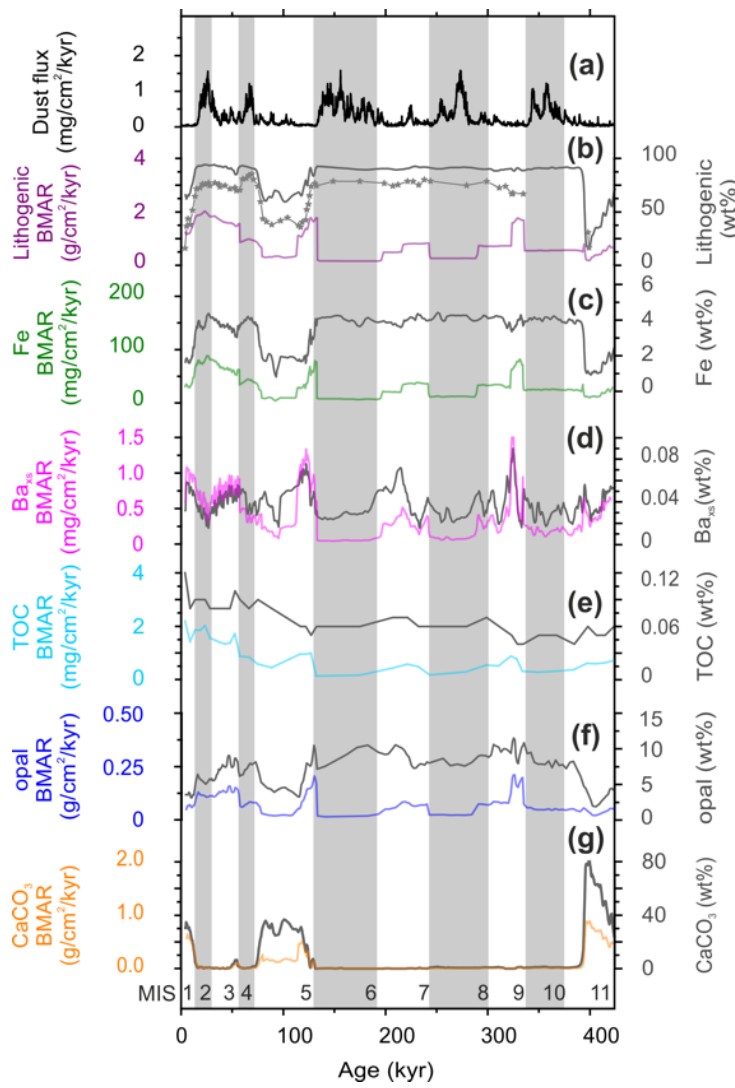

**Figure 2: SE Pacific (PS97/093-2) stratigraphy-based mass accumulation rate (BMAR) and percentages of individual components compared to dust flux in the EPICA Dome C (EDC) ice core (Lambert et al. 2008). (a) EDC dust flux; (b) Lithogenic content (wt. %; calculated from: 100-[opal wt. %+CaCO3 wt. %+2xTOC wt. %], grey line) and lithogenic content based on $^{232}$Th (grey stars); lithogenic BMAR (purple line); (c) Fe content (grey line) and Fe BMAR (green line); (d) TOC content (light grey line) and TOC BMAR (light blue line); (e) $Ba_{xs}$ content (grey line) and $Ba_{xs}$ BMAR (pink line);(f) Opal content (wt. %; grey line) and opal BMAR (blue line); (g) CaCO$_3$ content (grey line) and CaCO$_3$ BMAR (orange line). Numbers in the lower part of the figure indicate Marine Isotope Stages (MIS). Vertical grey bars mark glacial stages according to Lisiecki and Raymo (2005).**

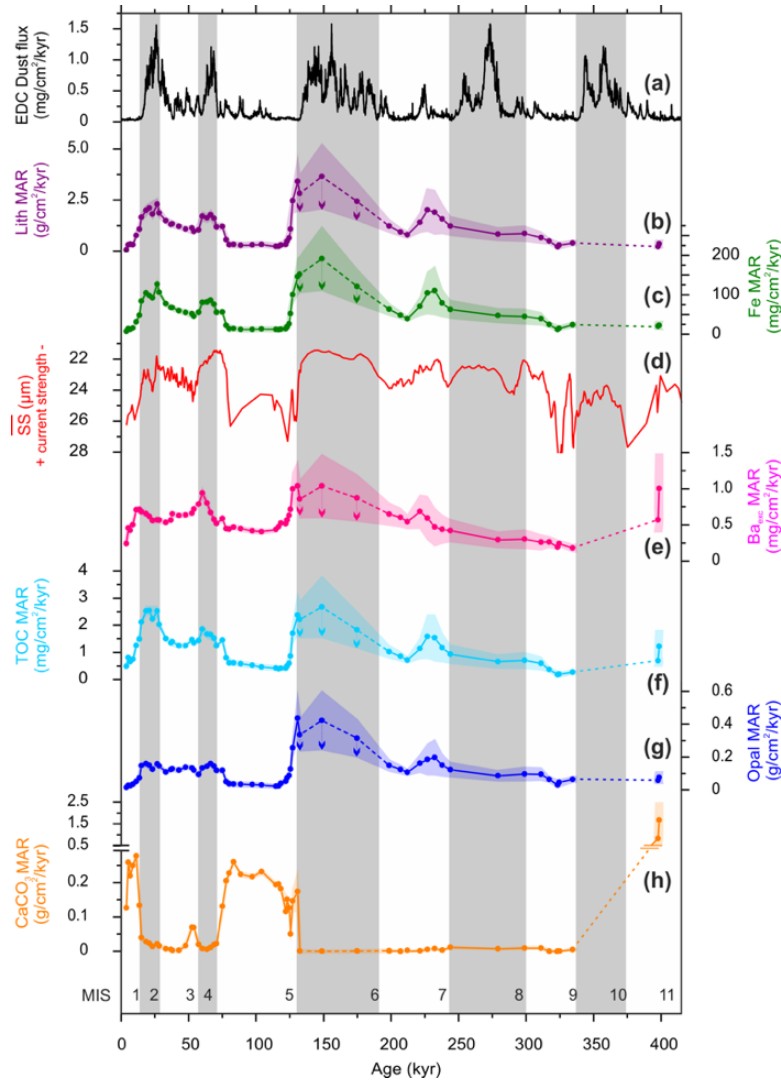


**Figure 3: Comparison of export production and terrigenous fluxes based on $^{230}$Th$_{xs}$-normalized mass accumulation rates (MAR) from SE Pacific core PS97/093-2 compared to dust flux in the EDC ice core (Lambert et al. 2008) and bottom current strengths (Toyos et al., 2020). (a) EDC dust flux, (b) $^{232}$Th based lithogenic MAR; (c) Fe MAR; (d) changes in bottom current strength as indicated by the sortable silt record of site PS97/093-2; (e) Ba$_{xs}$ MAR; (f) TOC MAR; (g) biogenic opal MAR; and (h) CaCO$_3$**

**MAR. Shaded areas indicate associated errors. Numbers in the lower part of the figure indicate Marine Isotope Stages (MIS). Grey bars denote glacial stages according to Lisiecki & Raymo (2005). Dashed lines during MIS 6 denote the interval of extreme winnowing where MAR might be overestimated, and for MIS 10 they indicate uncertainty due to the lack of data points.**


$^{230}\text{Th}_{xs}$-normalized mass accumulation rates (MAR) of individual components show that lithogenics, Fe, $\text{Ba}_{xs}$, TOC and opal

co-vary and display certain glacial-interglacial variability, whereas $CaCO_3$ peaks when almost all other fluxes are low (Fig 3). From MIS 7, lithogenic and Fe MARs show a pattern of higher values during glacials than in preceding interglacial stages and reach maxima during the MIS 6/5 transition. In contrast, the lowest lithogenic and Fe MARs are recorded in the Holocene, MIS 5, and possibly during MIS10-11 (Fig 3b, c). Opal MAR follows the same pattern with three strongest minima during MIS 11, MIS 5 and the Holocene. From mid-MIS 9 to the end of MIS 6, the opal MAR gradually increases, reaching the

highest values at the MIS 6/5 transition (Fig. 3g). Glacial-interglacial variability is not recorded between MIS 4 and MIS 2 in the opal record despite the relatively higher values. TOC and $\text{Ba}_{xs}$ MAR records are in good agreement with opal MAR except for the glacial peaks in the interval MIS 4 to MIS 2, and MIS11 (Fig. 3e, f). $CaCO_3$ MAR displays strong increases during MIS 11, MIS 5 and the Holocene only (Fig 3h).

Focusing factors range from significant winnowing ($\Psi=0.08$) to considerable focusing ($\Psi=3.55$) throughout the past 400 ka.

At the coring site, winnowing dominates, and we only have brief periods of focusing during the peak interglacials MIS 9, MIS 5e, and the Holocene. The intervals from MIS 11 to MIS 9 and from mid-MIS 8 to MIS 6 show net winnowing, whereas the interval between mid MIS 9 to mid MIS 8 was net neutral, and from MIS 5e to MIS 2 was nearly neutral with moderate winnowing (Fig 4b). Furthermore, the focusing factor record broadly agrees with the sedimentation rates, with winnowing in the intervals of low sedimentation rates and focusing during periods of relatively high sedimentation rates (Fig 4b, c).

A comparison between the MAR and the BMAR (Figs 4d, e) shows discrepancies between both records. For most of the record, MAR is higher than the BMAR, except for the focusing intervals (MIS 9, MIS 5e, and the Holocene), where an opposite trend is observed. The largest divergence between both records happens during MIS 6, which is characterized by significant winnowing (Fig 4). As a result, during MIS 9, MIS 6, MIS 5e and the Holocene, the BMAR and MAR of the individual components (lithogenic, Fe, $\text{Ba}_{xc}$, TOC, opal, and $CaCO_3$) diverge from each other to varying degrees (Table 1), showing only

similar values from mid-MIS 5 to MIS 2 and in MIS 9/8 (Figs 2 and 3).

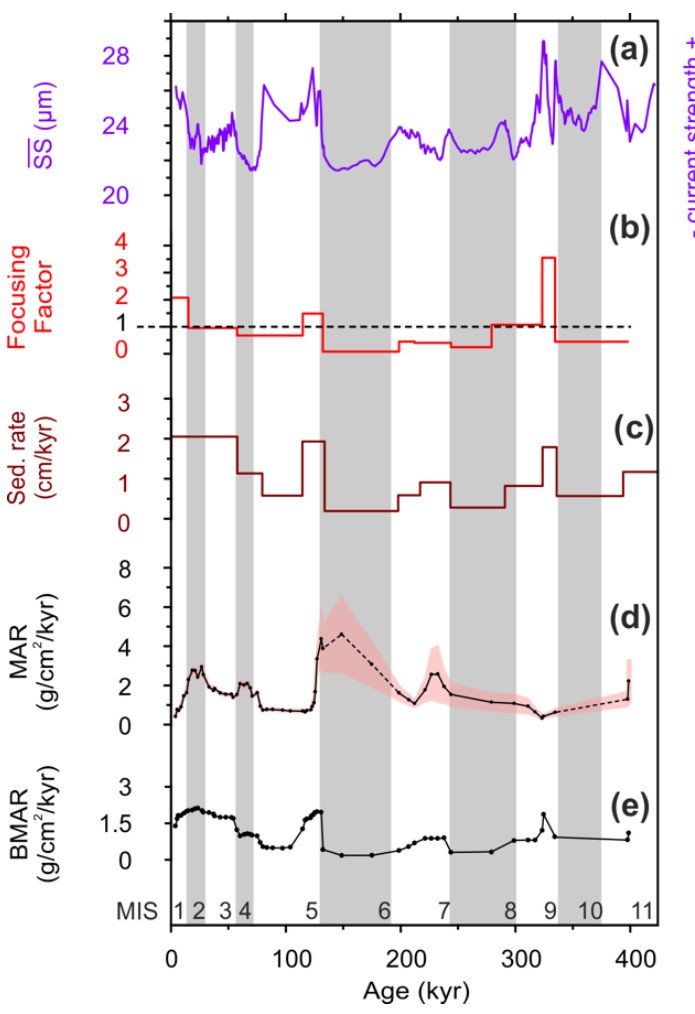

**Figure 4: Comparison of sediment mass accumulation rates, focusing factors, bottom current strengths and sedimentation rates of core PS97/093-2 for the last 400 ka. (a) Changes in bottom current strength based on the Sortable Silt grain size record (lilac line, Toyos et al., 2020) compared to (b) focusing factors (red line, black dashed line marks focusing factor=1); (c) sedimentation rates; (d) MAR ($^{230}Th_{xs}$-normalized, black line and dots) and associated errors (2σ; pink shadow), which grow with the age of the sample, dashed lines indicate intervals with uncertainties due to extreme winnowing (Marine Isotope Stage 6) or lack of data (Marine Isotope Stage 10); (e) BMAR obtained by multiplying sedimentation rates by dry bulk densities. Numbers in the lower part of the figure indicate Marine Isotope Stages (MIS). Grey bars denote glacial stages according to Lisiecki & Raymo (2005).**




**Table 1: Comparison between the average MAR and BMARs for the intervals with high (0–15 kyr, 324–334 kyr) and low (212–243 kyr, 243–279 kyr ) focusing factors.**

| intervals \ average | Accumulation rate of Fe (mg/cm²/kyr) | Accumulation rate of Ba$_{xs}$ (mg/cm²/kyr) | Accumulation rate of TOC (mg/cm²/kyr) | Accumulation rate of opal (mg/cm²/kyr) | Accumulation rate of CaCO$_3$ (mg/cm²/kyr) | |
|---|---|---|---|---|---|---|
| 0–15 kyr ($\Psi$=2.07) | 30.06 | 0.53 | 1.07 | 53.04 | 185 | MAR |
| | 40.65 | 0.94 | 1.82 | 70.05 | 411 | BMARs |
| | **1.35** | **1.77** | **1.70** | **1.32** | **2.21** | **overestimation factor** |
| 324–334 kyr ($\Psi$=3.55) | 55.44 | 0.46 | 0.59 | 125.46 | 11.50 | MAR |
| | 72.99 | 1.32 | 0.82 | 184.97 | 12.53 | BMARs |
| | **1.31** | **2.84** | **1.38** | **1.47** | **1.08** | **overestimation factor** |
| 212–243 kyr ($\Psi$=0.40) | 77.49 | 0.52 | 1.17 | 153.67 | 4.52 | MAR |
| | 34.21 | 0.30 | 0.50 | 69.14 | 1.40 | BMARs |
| | **2.26** | **1.71** | **2.35** | **2.22** | **3.21** | **underestimation factor** |
| 243–279 kyr ($\Psi$=0.24) | 47.11 | 0.29 | 0.66 | 86.05 | 6.62 | MAR |
| | 12.03 | 0.07 | 0.23 | 22.99 | 2.07 | BMARs |
| | **3.91** | **3.75** | **2.83** | **3.74** | **3.00** | **underestimation factor** |

## 5 Discussion

### 5.1 Influence of current dynamics on sediment redistribution and its effect on proxy record interpretations

The study area at the Pacific entrance of the DP is strongly influenced by the ACC, as documented by highly variable current strengths on millennial and glacial-interglacial timescales (Lamy et al., 2015; Toyos et al., 2020; Wu et al., 2021). Our results indicate that this variability in current strength influenced changes in sedimentation patterns (Fig. 4). Specifically, our focusing factors indicate that sediment redistribution is highly variable in our core, encompassing significant winnowing during MIS 6 (Fig 4). Under these extreme conditions winnowing may bias the [230]Th normalization reconstructions because of a preferential removal of the fine [230]Th-rich grains (Marcantonio et al., 2014). Costa and McManus (2017) documented a potential overestimation of MAR when winnowing is severe (focusing factors of 0.158–0.20). With moderate winnowing $\Psi$>0.24, this effect is negligible and [230]Th$_{xs}$-normalized accumulation rates are robust. Therefore, in core PS97/093-2, the high MAR during MIS 6 may in part stem from the extreme winnowing ($\Psi$=0.08) at that time, whereas in all other intervals $\Psi$ exceeds 0.24 and thus estimated MARs are reliable (Fig 4).

In our core, we observe winnowing during glacial periods in intervals with relatively low current strengths, and increased focusing factors contemporaneous with increased sortable silt values in the same core, reflecting strengthened bottom currents (Toyos et al., 2020; Fig. 4). The consistent pattern of winnowing (focusing) in intervals with relatively low (high) current strengths might at first sight seem counterintuitive. Since under strong bottom current conditions, the fine sediment fraction





(<10 µm) potentially behaves similarly to the >63µm sediment fraction because of cohesive effects and flocculation of the fine fraction during transport (McCave and Hall, 2006), but under slower current conditions a loss of such effects in the fine sediment fraction might trigger the winnowing of the fine fraction (Marcantonio et al., 2014). We suggest that winnowing at our site, in a generally fine-grained sediment, is due to current velocities that occasionally are overruling the cohesive forces,

resulting in a loss of a small amount of the fine fraction. Prior studies have pointed out that in the deep sea, current velocities between *ca.* 6.5 to 10.5 cm/s are enough to trigger surface sediment erosion of the aggregated particulate matter that compose the surface sediments (Peine et al., 2009; Turnewitsch et al., 2008). However, our results suggest that even lower flow speeds of ~5.5–6 cm/s during glacials (Fig 4, Toyos et al., 2020, calibrated following Wu et al., 2019) reached the threshold of intermittent erosion and resuspension of the fine sediment fraction. During interglacial periods MIS 9, MIS 5 and the Holocene,

strong currents plausibly cause a gradual mobilization of more and more coarse-grained material that was laterally accumulated in certain areas, resulting in a coarser-grained focusing at our site. A similar pattern, characterized by strengthened bottom currents inferred from grain size analysis, and increased sediment focusing has also been observed for interglacials on the Weddell Sea continental margin (Frank et al., 1996).

It is evident that the dynamic bottom water circulation led to frequent syndepositional redistribution. For core PS97/093-2,

when comparing MAR and BMAR of Fe, $Ba_{xs}$, TOC, opal and $CaCO_3$ in the intervals with high and low focusing factors (excluding MIS 6, where we cannot discard an overestimation of the $^{230}$Th fluxes because of extreme winnowing), large discrepancies are seen (Table 1, Figs 2 and 3). BMAR of the individual components are up to *ca.* three times higher than MAR in intervals with highest focusing, and *ca.* four times smaller in intervals with pronounced winnowing (Table 1). These differences confirm that the PS97/093-2 BMAR are strongly biased by sediment redistribution due to current dynamics. Given

that the PS97/93 site is affected by substantial lateral redistribution of sediment particles, we therefore base our paleoproductivity reconstruction on the MAR, which allows for the quantification of lateral sediment redistribution and accurate vertical rain rates (e.g., Costa et al., 2020; Suman and Bacon, 1989).

**5.2 SE Pacific lithogenic material, sources and iron fertilization potential**

A comparison of lithogenic fluxes between PS97/093-02 core with other records from open ocean sites in the SAZ and in

Antarctic ice cores, interpreted as eolian dust (Anderson et al., 2014; Lambert et al., 2008; Lamy et al., 2014; Martínez-Garcia et al., 2009; Thöle et al., 2019), shows a similar pattern in lithogenic MAR fluctuations (Fig. 5). However, for MIS 10 and MIS 8, some characteristic features are not observed in our sediment core due to a lower temporal resolution that prevents an accurate evaluation during such intervals (Fig 5). Our glacial lithogenic MAR peaks were one order of magnitude higher than those reconstructions from the open Pacific, Atlantic and Indian SAZ (Fig 5). Furthermore, studies of modern surface

sediments discard a substantial contribution of dust to the DP region and the SE Pacific north of the SAF (Wengler et al., 2019; Wu et al., 2019). Thus, an additional non-eolian source of terrigenous material is required.

We suggest that the comparably high lithogenic MAR are due to the relative proximity of our site to southern South America as a substantial source of fluvially-derived and glaciogenic sediments, particularly during glacials when large parts of the





southern Andes were covered by the PIS (Fig 1b). From MIS 6 onwards, PS97/093-2 lithogenic MAR reveal significant

increases during glacials. Specifically, the most intense growth occurred during MIS 6, due to a combination of the penultimate local glaciation (Rabassa, 2008) and a possible winnowing bias, followed by a smaller rise during MIS 4, and a gradual increase to higher values in the LGM. This pattern closely corresponds to reported Patagonian ice expansions (e.g., Gowan et al., 2021; Lowell et al., 1995; Rabassa, 2008; Rabassa and Clapperton, 1990; Fig S2). Furthermore, an IRD record from Core MD07-3128, retrieved from the Pacific entrance to the Strait of Magellan (*ca.* 620 km NNW of core PS97/093-2, Fig 1b), shows a

consistent pattern of higher IRD contents during cold intervals over the past ~60 kyr, which have been interpreted as advances of the PIS (Caniupán et al., 2011). Moreover, enhanced Fe concentrations linked to PIS advances during colder periods were reported at ODP Site 1233 from the Chilean continental margin (Fig 1b, Kaiser and Lamy, 2010; Lamy et al., 2004), and strong similarities between the dust peaks in Antarctica and the advances of the PIS have been reported by Sugden et al. (2009).

It has been suggested that the Fe bio-availability depends on the mineralogy of the detrital fraction (Schroth et al., 2009).

Therefore, it is unclear which amount of the Fe that reached our site was bioavailable, and how this availability changed through time. Previous studies suggest that physical weathering increases the labile primary Fe (II) content of sediments that is associated with phytoplankton fertilization (Shoenfelt et al., 2017, 2018), indicating that a stronger contribution of glaciogenic sediments increases the proportion of bioavailable Fe in a given sediment flux (Shoenfelt et al., 2019). Nevertheless, since the study site is proximal to the continental sources (tip of South America), which are dominated by

physical weathering during both glacial and interglacial intervals, we hypothesize that temporal changes in the proportion of bioavailable Fe in the SE Pacific are less pronounced than in more open ocean locations, since we expect that the Fe supply delivered to our site might have always been relatively enriched in bio-available Fe compared to other oceanographic sectors. In summary, because dust does not seem to be the main source of lithogenic material at site PS97/093, we propose that advances of the PIS and iceberg discharge during glacial intervals (e.g., Kaiser et al., 2007; Rabassa, 2008) could have increased

lithogenic fluxes and potentially bio-available Fe reaching the subantarctic Pacific entrance to the DP and thus promoting biological productivity.

In the present-day Atlantic SAZ, chlorophyll blooms are supported by an advective supply of Fe from the South American coastline along the western boundary current first and the Subtropical Front later, rather than by continental dust sources (Graham et al., 2015). Other studies also invoke this mechanism for the LGM, where increased supply of glaciogenic debris

by expanded Patagonian and Antarctic Peninsula ice sheets travel long distances via the ACC at depth (Noble et al., 2012). However, in the subantarctic region of our core location, the glacial northward shift of the SAF likely decreased the DP throughflow of the ACC, weakening the cold water route into the Atlantic ( Lamy et al., 2015; Toyos et al., 2020), which would have hindered the transport of Fe released from the PIS to the Southeast Pacific into the Atlantic. It has been suggested that the reduced supply to the Atlantic basin through the DP during glacials might have been compensated by a stronger

recirculation within the South Pacific, causing a stronger South Pacific Gyre re-circulation (Lamy et al., 2015). Therefore, the bioavailable Fe might have been transported in the South Pacific gyre, enhancing productivity in this region, which is currently limited by Fe availability (Bonnet et al., 2008). To evaluate the reach of Fe fertilization in the SE Pacific linked to PIS





dynamics, further reconstructions along the Pacific SAZ and Cape Horn-Humboldt current systems are needed. Currently, the knowledge is restricted to one recent record (core GeoB3327-5, located at 43ºS, *ca.* 400 km off the Chilean coast) that

reportedly featured increases in primary production during MIS 2–4, linked to increased supply of micronutrients via continental runoff, primarily controlled by PIS variability or precipitation (Tapia et al., 2021).

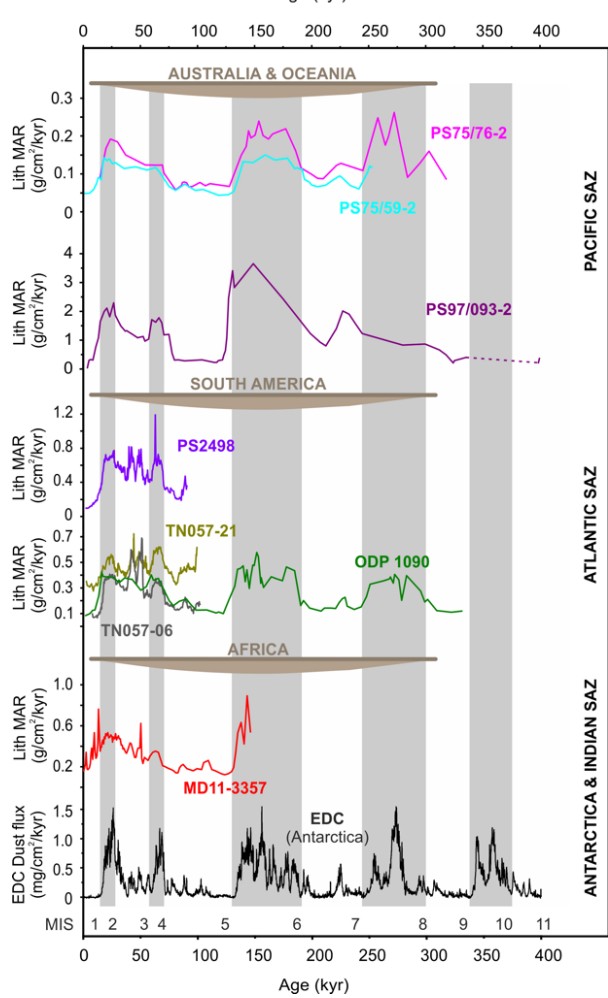

**Figure 5: Changes in [230]Th-normalized Lithogenic MAR in the subantarctic Southern Ocean (see Fig 1 for location of sites discussed). From top to bottom: Cores PS75/076-2 and PS75/059-2 (Lamy et al., 2014); Core PS97/093-2 (this study); Core PS2498 (Anderson**

**et al., 2014); Cores TN057-21 and TN056-06 (Anderson et al., 2014), and ODP1090 (Martínez-Garcia et al., 2009); Core MD11-3357 (Thöle et al, 2019), and Dust MAR in the EPICA Dome C ice core (Lambert et al., 2008). Marine Isotope Stage (MIS); grey bars denote glacial stages. Continental masses are located in their relative longitudinal position to the marine sediment cores and represent the primary source of terrigenous inputs to the Subantarctic Zone.**





### 5.3 Orbital-scale variations and drivers of export production in the SE Pacific SO

**5.3.1 Assessing the proxies' response to export production fluxes**

Our $^{230}$Th-normalized results show that the proxies for integrated export production (Ba$_{xs}$ MAR and TOC MAR) generally co-vary and that both also vary in parallel and show significant positive correlations with biogenic opal MAR; all being lower during full interglacials (Holocene, MIS 5 and MIS 11; PAGES, 2016) and higher during MIS 6 and MIS 4–2 (Table S1, Fig 3).

Usually, variable dissolution in the water column does not significantly affect opal burial in the SO (Chase et al., 2015), suggesting that opal MAR might be only marginally affected by preservation changes, thus providing valid information on past diatom productivity changes (e.g., Bradtmiller et al., 2009; Sprenk et al., 2013). Since the preservation of organic carbon in sediments is globally scarce (about 1%, Berger et al., 1989), the good correspondence and positive correlation between opal and TOC MAR (r$^2$=0.56, Table S1, Fig 3) might imply that the TOC preserved in the sediments is related to carbon locked in

diatom frustules. Opal MAR also correlates with Ba$_{xs}$ MAR, which has a preservation rate of *ca.* 30% in sediments under oxic conditions (Dymond, 1992), suggesting that most of the glacial export production was fueled by diatoms. An exception to the strong correlation between the export production proxies occurred from MIS 4 to MIS 2, when TOC and Ba$_{xs}$ MARs show relatively high values with glacial/interglacial variability, whereas the opal record is rather flat within this interval. Several studies proposed that diatom growth inferred from opal burial during the LGM may have been more effective than today,

because diatoms can reduce their Si/C uptake ratio under Fe-replete conditions (Anderson et al., 2002; Chase et al., 2015). Thus, a glacial increase of Fe input may have created conditions under which relatively more TOC and biogenic barium were exported per unit of opal buried (Fig. 3).

Unlike our other MAR records, the highest CaCO$_3$ MAR values are observed only during the Holocene, MIS 5 and MIS 11 (Fig. 3). To explain the different pattern, we have to consider changes in carbonate chemistry and variations in the depth of

the lysocline affecting our site's carbonate contents through time. At present, the lysocline occurs at *ca.* 4000 m (Sulpis et al., 2018), which is only slightly below the depth of our core (3781 m water depth). Site PS97/093 is bathed by Circumpolar Deep Water, which is characterized by relatively low [CO$_3^{2-}$], being undersaturated with respect to CaCO$_3$, and therefore promotes carbonate dissolution (Key et al., 2004). In the SO, the lysocline depth shoaled at least 0.5 km during the glacial stages of the past 500 kyr, triggering enhanced carbonate dissolution and concomitant reduced carbonate MAR (Howard and Prell, 1994).

Moreover, poor carbonate preservation has also been observed in another SE Pacific core with an alkenone-based SST record (PS75/034-2, 54º 22'S, 80º 05'W, 4425 m water depth; Ho et al., 2012, Fig 1b). Nevertheless, since carbonate preservation is favored in intervals with high sedimentation rates (Gottschalk et al., 2018), in our core the relatively high sedimentation rates during MIS1–3 or MIS 5e (Fig 4c) might allow the burial of carbonates in the sediment before being dissolved during such intervals. On the other hand, we suggest that the absence of carbonates between MIS 10 to MIS 6, and in MIS 4 indicates

carbonate dissolution, rather than changes in export production of coccolithophores and foraminifera (Fig. 3).



### 5.3.2 Drivers of local export production patterns

In the DP, the fronts act as barriers to mixing in the ocean, preventing cross-frontal movement and meridional exchange of water masses and their associated nutrients (Naveira Garabato et al., 2011; Paparazzo, 2016). On longer timescales, studies of sediment cores located upstream and downstream of the northern and central DP suggest a northward shift of the SO frontal system during glacial times, lower ACC flow speed and reduced transport through the DP (Lamy et al., 2015; Roberts et al., 2017; Toyos et al., 2020; Wu et al., 2021). Particularly at site PS97/093, a comparison between the $\overline{SS}$ record, and the opal and TOC MAR records shows that decreases in current strengths correspond with increases in opal and TOC (Fig 3). The inverse correspondence between current strength and opal export supports the idea that the glacial decrease in current vigor, linked to the northward shift of the frontal systems, would locate our coring site south of the strongest ACC flow in the vicinity of the SAF (Toyos et al., 2020) and closer to the Si-rich southern waters. Furthermore, it is likely that the waters between the PF and SAF received an extra supply of dissolved $Si(OH)_4$ during glacial intervals as inferred from the displacement of the opal belt north of the present day PF position (e.g., Diekmann, 2007). This pattern is explained by upwelling of dissolved $Si(OH)_4$ south of the PF as it does today, but due to expanded ice cover during glacials, very little silica was used by diatoms south of the PF, and the upwelled $Si(OH)_4$ was transported northwards from the PF (Chase et al., 2003; Freeman et al., 2018). At the same time, we discard a substantial arrival at our core site of either hydrothermally sourced Fe from the East Pacific Rise (Fitzsimmons et al., 2014) or potentially upwelled Fe south of the PF, because of in the hypothetical case of upwelling of Fe south of the PF, its strong particle reactivity might have caused efficient scavenging of Fe beneath the ice (Chase et al., 2003). Therefore, enhanced nutrient utilization and opal production can only be achieved through an additional Fe source. In our core, the coeval increase in MAR of lithogenics, Fe, biogenic opal, TOC and $Ba_{xs}$ suggests that sufficient Fe was transported to our site to consume all the supplied $Si(OH)_4$ during glacial intervals. Furthermore, under Fe-replete conditions, diatoms take up substantially less silicic acid relative to nitrate (Matsumoto et al., 2014). Modern observations and incubation experiments show that the Si:N uptake ratio lowers from ~ 4:1 to ~ 1:1 (Brzezinski et al., 2002; Franck et al., 2000). Therefore, since our core received enough glacial Fe to relief the limitation, the enhanced diatom production, indicated by higher biogenic opal MAR, may respond to a reduced diatom Si:N uptake ratio, combined with an increase of the amount of $Si(OH)_4$ that reached the core site.

In contrast with this glacial scenario, increases in bottom current strength, usually during interglacials, indicate that the SAF was located south of our site, resulting in a production regime similar to present-day conditions. This area is characterized by low average chlorophyll-*a* concentrations (Fig 1b), where the lack of dissolved silica may limit the growth of diatoms (Demidov et al., 2011; Freeman et al., 2019), Fig 6a and 6b) and might cause coccolithophores and other phytoplankters to become the dominant group (Rigual Hernández et al., 2020; Saavedra-Pellitero et al., 2019). However, the absence of peaks in the TOC and $Ba_{xs}$ MAR during the Holocene and MIS 5 suggest that the increased $CaCO_3$ export production did not impact significantly in the integrated proxies of export production, implying that export production by calcareous organisms was too small to have a noticeable impact on the total export during such intervals (Fig 3).



Unlike other interglacials, during MIS 11, a prominent nannofossil ooze, primarily composed of *Gephyrocapsa*
coccolithophore and foraminifera has been reported in the area (Lamy, 2016; Toyos et al., 2020) that is mirrored at our site by
$CaCO_3$ MARs one order of magnitude higher than in the Holocene and in MIS 5, and by increases in the $Ba_{xs}$ and TOC MARs
(Fig 3). A similar rise in $CaCO_3$ accumulation has also been recognized at other SAZ locations in the Atlantic (Hodell et al.,
2000) and the Pacific (Gersonde, 2011; Saavedra-Pellitero et al., 2017) sectors of the SO, suggesting that the increase in
PS97/093-2's $CaCO_3$ MAR during MIS 11 is part of a general pattern comprising at least the Pacific and Atlantic SAZ, likely
caused by an exceptional southward migration of the ACC frontal systems in the Pacific sector of the SO (Saavedra-Pellitero
et al., 2017).

### 5.3.3 Comparison of PS97/093-2 export fluxes/MARs to other Southern Ocean SAZ sites

In order to assess the spatially complex, zonal dynamics of productivity and lithogenic flux variations along the Southern
Ocean's SAZ, we compare our MAR reconstructions with previously published records from open ocean areas of the SAZ
(Anderson et al., 2014; Frank et al., 1999; Lamy et al., 2014; Martínez-Garcia et al., 2014; Nürnberg et al., 1997; Thöle et al.,
2019). We use a circum-Antarctic subantarctic transect of MAR of lithogenics, Fe, $Ba_{xs}$, TOC, opal and $CaCO_3$ for the
Holocene (0–10 ka), the LGM (19–27 ka; Clark et al., 2009), the last interglacial MIS 5e (119–124 ka), and the end of MIS 6
during full glacial conditions, just before the beginning of the warming in the SO at 132–131 ka (Bianchi and Gersonde, 2002;
Fig 6). Overall, our productivity indicators agree with subantarctic records from other locations, displaying higher $CaCO_3$
during interglacials and higher opal, TOC and $Ba_{xs}$ during glacials, the latter pattern most likely enabled by an increase of
lithogenic Fe fluxes (Martin, 1990; Fig 6). But, as it has previously been reported, it is likely that other factors than Fe solely
regulate the biological production in the SO (Anderson et al., 2014; Kohfeld et al., 2005).
At our core location, we observe that the LGM lithogenic (Fe) MAR were about eightfold (ninefold) larger than during the
Holocene. Furthermore, our site records the highest glacial lithogenic and Fe fluxes of the subantarctic SO; the LGM Indian–
Atlantic and Pacific lithogenic fluxes are four to eleven times lower than at PS97/093 site (Fig 6). However, the corresponding
glacial increases in our productivity MARs are not as high as expected when considering the relative increase in the lithogenic
MAR. In the Atlantic and Indian sectors, the LGM rise in export production was larger than the increase in lithogenic fluxes,
whereas the opposite pattern is observed in our core (Fig 6). Furthermore, the glacial and interglacial export production fluxes
are higher in the Atlantic and Indian sectors than at the DP entrance and in the central Pacific (Fig 6). Whereas in the central
Pacific, the relatively low export production is explained by weaker Fe fertilization caused by lower local glacial dust fluxes
(Lamy et al., 2014), the results at our site indicate that export production did not respond to glacial Fe fertilization as efficiently
as in the other sectors of the SAZ (Fig 6).Therefore, we suggest that under bioavailable Fe-replete conditions, another
mechanism than Fe fertilization might ultimately regulate export production at our site.

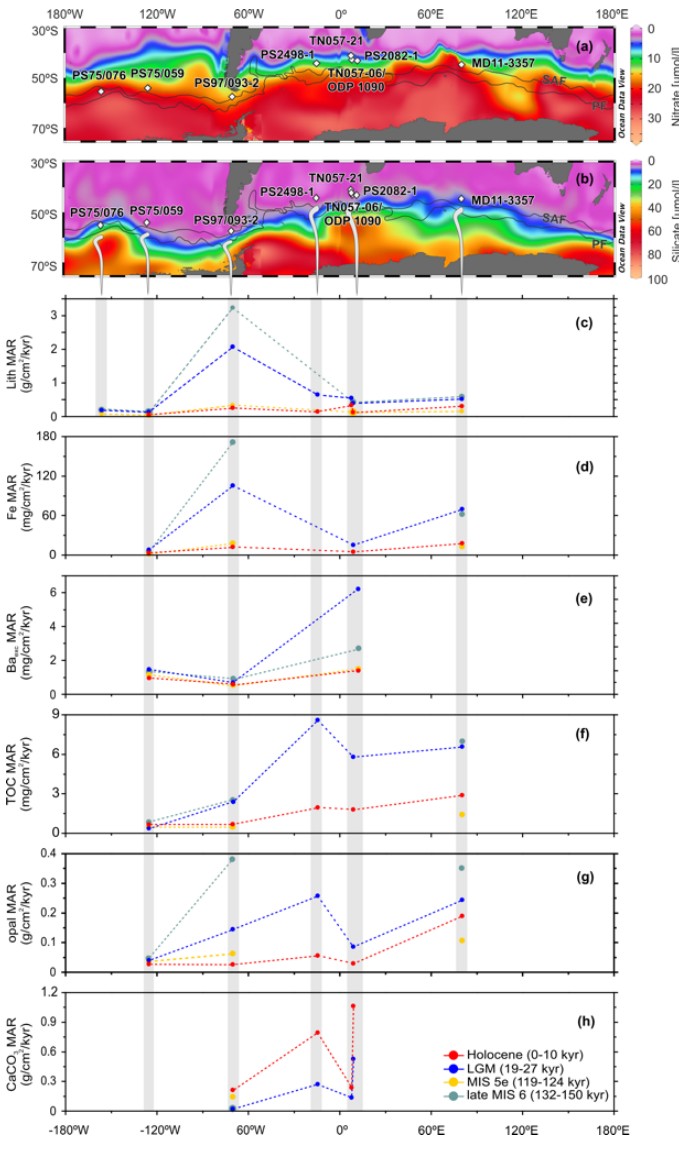

**Figure 6: Comparison of MAR across the Subantarctic Southern Ocean, during the Holocene (0–10 ka, red), the Last Glacial Maximun (19–27 ka, blue), Marine Isotope Stage 5e (119–124 ka, yellow), and late Marine Isotope Stage 6 (132–150 ka, green): Maps of surface water nitrate (a) and silicate (b) concentrations (data from WOCE, Global Hydrographic Climatology, Gouretski and Koltermann, 2004); black lines indicate the modern location of the Subantarctic Front (SAF), and Polar Front (PF, Orsi et al., 1995); white diamonds indicate the core locations (PS75/076 and PS75/059, Lamy et al. 2014; PS97/093-2, this study; PS2498-1, TN057-21 and TN057-06, Anderson et al., 2014; ODP1090, Martínez-García et al. 2014; PS2082-1, Nürnberg, 1997 and Frank, 2002; MD11-3357, Thöle et al., 2019); (c) lithogenic flux; (d) Fe flux; (e) Ba_xs flux (f) Total organic carbon flux; (g) Opal flux. (h) Carbonate flux. Grey bars indicate the projected core positions.**





The region is currently nitrate and phosphate-rich (Fig 6a; phosphate shown in Fig S3), and previous works north of the PF have shown that productivity increased during the LGM, whereas nitrate utilization decreased only slightly, implying a glacial

intensification in the supply of nitrate to surface waters (François et al., 1997; Martínez-Garcia et al., 2014). Since nitrate is not the macronutrient that was limiting export production during glacials, we propose $Si(OH)_4$ as a suitable candidate.

Under high Fe conditions, a glacial northward migration of the SO frontal system, together with the transport of silicic acid northwards of the PF (Chase et al., 2003), should have contributed to the relief of silicate limitation in the SAZ, resulting in an increase in export production (predominantly diatom production). Given that LGM (and the possibly overestimated MIS 6)

opal fluxes at our site were over fivefold larger than during the Holocene (Fig 6g), to sustain such a diatom productivity implicated by the opal flux would require a supply of $Si(OH)_4$ much greater than exists today (Anderson et al., 2014). Thus, we propose that if all the $Si(OH)_4$ brought to our site was consumed entirely, silica (diatom) production would have been inhibited by $Si(OH)_4$ limitation, as it is happening today in some areas of the Polar Front Zone, where nearly depleted levels of silicic acid have been reported ( Mengelt et al., 2001, Fig 6b). This hypothesis might be supported by modern experiments

that show that $Si(OH)_4$ limitation significantly restrict the response of diatom production to Fe in the SAZ (Brzezinski et al., 2005). Consequently, the growth of other phytoplankton groups (e.g., small flagellates) or small, weakly-silicified diatom species would have been favored, which are strongly affected by remineralization, thus lowering export production (Buesseler, 1998). Alternatively, since increases in productivity at the SO fronts are explained by horizontal advection rather than upwelling (Graham et al., 2015), it might be possible that during past glacials only a portion of the $Si(OH)_4$ supplied via

upwelling at the PF travelled northwards across the PFZ reaching the core location. The other portion of the upwelled silica might have been advected laterally into the Atlantic via the ACC (Graham et al., 2015), causing a more intensive phytoplankton growth there instead of in the DP. Although the latter scenario is somewhat speculative, both assumptions (a moderate glacial $Si(OH)_4$ limitation and dominance of horizontal advection rather than upwelling) are not mutually exclusive; hence the two scenarios would explain the weaker productivity response to Fe at the PS97/093 site compared to other subantarctic records.

**6 Conclusions**

We present a multi-proxy approach for reconstructing paleo-export production in the SAZ at the Pacific entrance to the DP covering the past 400 ka. As our core is located in an area with relatively high bottom current speeds, we show that our site is strongly affected by the lateral redistribution of sediments, with high focusing (winnowing) factors associated with high (low) bottom current speeds. This counterintuitive pattern is most likely due to a dominance of the coarse (fine)-grained fraction

together with a preservation (loss) of the flocculation and cohesive effects of the fine sediment fraction under relatively high (slow) bottom currents. As a result, focusing (winnowing) leads to an overestimation (underestimation) of the BMAR by a factor up to ca. 3 (4) during such intervals in our core. The frequent syndepositional redistribution emphasizes the importance of the utilization of the $^{230}Th_{xs}$-normalization method to reconstruct fluxes in the subantarctic SO.

Whereas our lithogenic record follows the temporal pattern of other lithogenic records at open ocean sites in the SAZ and in the Antarctic ice core (EDC dust flux), we present evidence that dust was not the main source of lithogenic material. Instead, advances of the PIS during glacial intervals increased the portion of glaciogenic sediments and therefore released lithogenics. Since the core site is relatively proximal to continental sources, which are dominated by physical weathering and likely enriched in bioavailable Fe, we suggest that the SE Pacific (Drake Passage) received enough glacial bioavailable Fe to not limit phytoplankton growth.

Our results show orbital-scale variations from a predominantly high opal-rich export production for most glacials to lower levels of export productivity during interglacials dominated by carbonates. We suggest that such fluctuations are responding to glacial-interglacial frontal shifts coupled with nutrient limitation. We hypothesize that during interglacials the SAF was located slightly south of our core location, so that a reduction in Fe and dissolved silica supply may have limited the growth of siliceous phytoplankton (diatoms), leading to i.e. coccolithophores becoming the dominant group. Yet, the reduced total 550 export production during such intervals suggests that export production by calcareous organisms was too small to have a noticeable impact on the total export, except for MIS 11, where the impact was perceptible due to a prominent calcareous ooze. Conversely, the glacial northward migration of the SO frontal system would have positioned the core south of the SAF and closer to Si-rich waters in an environment characterized by extremely high lithogenic and Fe fluxes from a more extensive PIS as mentioned above. The coeval increase in lithogenic, Fe, opal, TOC, and $Ba_{xs}$ suggests that our core received sufficient Fe 555 and $Si(OH)_4$ to fuel higher export production during glacial intervals.

A spatial comparison with previously published records from different locations in the SAZ shows that our core displays the highest glacial lithogenic and Fe MARs. However, the corresponding glacial increases in productivity fluxes are not as high as we expected given the relative increase in lithogenics, suggesting that export production did not respond to glacial Fe fertilization as efficiently as in other sectors of the SAZ. We hypothesize that under these Fe-repleted glacial conditions, a 560 depletion in the silicic acid that was transported northwards of the PF, could explain the glacial export production pattern at our core location.

**Data availability**

All data are available in PANGAEA repository: (no DOI assigned yet)

**Supplement**

The supplement related to this article is available online at: XXXX

**Author contribution**

This study is part of MHT's thesis under the supervision of CBL and FL. MHT carried out most of the measurements. MHT, GW and HWA made the calculations. MHT, CBL, FL, HWA, LLJ and GW analysed and interpreted the data. MHT wrote the first draft and produced the figures for the manuscript with substantial contributions of GW, HWA, LLJ, CBL and FL. All
authors interpreted, edited, and reviewed the manuscript.

**Competing interests**

The authors declare that they have no conflict of interest.

**Acknowledgements**

This work was funded by the Alfred-Wegener-Institut Helmholtz-Zentrum für Polar- und Meeresforschung through its research
programmes "PACES-II" and "Changing Earth – Sustaining our Future". Additional support was provided by the Chilean oceanographic centers FONDAP-IDEAL (project number 1500003) and COPAS Sur-Austral (AFB170006) to CBL and MHT, and Lamont-Doherty Earth Observatory. MHT acknowledges support from scholarship CONICYT-PCHA/Doctorado Nacional/2016-21160454, Postgraduate Office of Universidad de Concepción, Red Clima Red (project number LPR163), and Doctorado MaReA. We thank the captain, crew and scientific party of R/V Polarstern for a successful PS97 cruise. We
acknowledge A. Ávila for technical support at the Paleoceanography lab of Universidad de Concepción, R. Schwarz, M. Fleisher and J. Abell for assistance at Lamont-Doherty Earth Observatory, and S. Wiebe, R. Fröhlking and V. Schumacher for technical support at AWI.

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
