# Peer review of "Variations in export production, lithogenic sediment transport and iron fertilization in the Pacific sector of the Drake Passage over the past 400 ka"

_Climate of the Past, 2021_

## Referee Comment (RC1)

**General Comments**

This manuscript presents new data from a sediment core located close to the Drake Passage. These data allow the authors to discuss lithogenic fluxes to their core site and their influence (as bioavailable Fe) in shaping the character and magnitude of export production in the Subantarctic region of the Southern Ocean, over multiple glacial-interglacial cycles. One innovative component is the fact that all their Mass Accumulation Rates are Th-corrected, which is a great practice as it takes into account focussing/winnowing effects, besides allowing the authors to provide insights on current dynamics, when coupled, as they do, with other methods (i.e., sortable silts). Another important aspect of this study is the comparison to a host of other records available from the different Sectors of the Southern Ocean, allowing the reader to get a circum-antarctic perspective of export production variability through time. Methods are valid and results presented in a very clear, easy to follow style, with appropriate referencing of related work. The paper is nicely written, logically structured and figures and tables are both of good quality and appropriate.

**Specific Comments**

Lines 44-46: Fronts as barriers: In principle, yes, but mesoscale dynamics/eddy fields allow a certain level of mixing across fronts.

Lines 159-160: "For tuning, we assumed that low Fe contents characterize interglacial periods, whereas high contents represent glacials". Given how this paper is specifically addressing issues related to Fe-depleted/Fe-replete conditions, it would be good to provide a bit more information on how much (or how little) the tuning distorts the preliminary age model. Or, at least, a more in-depth mention of the supporting evidence for this choice (rationale behind it, and support from other measurements, like carbonate, Ca counts, etc).

Lines 176-177: "For terrigenous contents exceeding 75%, opal concentrations measured at UdeC are consistently 3–5% higher than those measured at AWI". Even if the subsequent mention of similar patterns of variability is right on point, it would be useful to mention the potential reasons for such systematic difference.

Line 208: Does the relatively lower match (compared to Fe and Ca) between XRF and bulk sediment measurements for Ba relate entirely to the two main mentioned uncertainties (Ba as marine barite and assumed constant Ba/Ti ratio for terrigenous components), or are there other factors potentially affecting this deviation?

Line 246: Ba excess higher during interglacials: interestingly, mainly true for MIS 5, 7, 9, not so much for either MIS 1 or 11… So that peculiar pattern might have a reason behind it.

Line 250: Carbonate and Ba excess BMAR varying in parallel to their respective percentages. True in general, but different pattern (for both of them) during MIS5e compared to MIS 5a-d. Which, again, might be a peculiarity worth exploring.

Legend to Figure 3: The mention of export production is a bit misleading: these are burial rates in the sediment, as derived by MARs…. By their nature, compared to export production, they integrate the effect of dissolution and remobilization (focussing/winnowing) through the water column and at the water/sediment interface. In the same legend: how the error envelopes were derived should be mentioned somewhere in the text under Methods.

Lines 276 and 279: From MIS 7… From mid-MIS 9… To a certain extent, both of these expressions do not really indicate the timing of initiation of a clear pattern but have more to do with the availability of data dense enough as to recognize that pattern in the first place.

Line 368: "and strong similarities between the dust peaks in Antarctica and the advances of the PIS". Just as you did for the IRD record, mention explicitly why this observation is relevant to the glaciation status of PIS and its role in increased lithogenic material input to your core site during glacials. You mentioned this in general terms in lines 358-359. This is an important point to clarify also in light of your statements on the EDC dust record being mainly linked to eolian inputs… essentially re-iterating that enhanced glacial conditions in PIS lead to increased terrestrial input to the adjoining ocean. Or you may decide that it is not necessary to repeat this again here, as you describe exactly how this this interplay works when talking about bioavailable iron in lines 378-380.

Line 408: "all being lower during full interglacials (Holocene, MIS 5 and MIS 11". Two remarks: You do provide a PAGES statement for this statement, but usually MIS 7 and 9 are considered to be full interglacials as well… Maybe one way to avoid the diatribe of what is a full interglacial and what is not, one could rephrase this to "being lower during some interglacials…". Second remark: the patterns are difficult to argue for MIS 11 as based on just 2 datapoints… While the statement might still be ok for Holocene and MIS 5, it definitely isn't true for Baxs during MIS 11, where the measured value is actually the highest on your record.

Lines 412-413: "Since the preservation of organic carbon in sediments is globally scarce (about 1%". Please specify what this 1% refers to: is it the percentage of TOC produced in the water column that gets preserved in sediments, or is it (a sort of globally-averaged) percent content of TOC in sediments? I guess the former, but not sure.

Line 419: "diatom growth". Not 100% sure growth is the right word here… could "diatom frustule multiplication" or "diatom blooms" be better? Also, I think the sentence does not read right… suggestion "… that diatom frustule multiplication was more effective during the LGM compared to today (resulting in higher opal burial during the LGM), as diatoms reduce their Si/C…"

Lines 434-435: In connection to what discussed in my comment to line 408 (your/PAGES statement about "full interglacials", excluding MIS 7 and 9), it would probably be interesting to mention how those two interglacials did not seem to manage to "offset" carbonate
dissolution as all other interglacials did in your record…

Lines 512-516: Besides the mechanisms you propose in this section, an additional one could be mentioned (besides frontal, and accompanying nutrient fields, movements and increased Fe supply) as playing a role in pushing/supporting the glacial system towards increased opal productivity during glacials: the better Si/N utilization by diatoms under Fe-replete conditions (that you mentioned elsewhere, with references to Brzezinski&al and Frank&al), which would also provide a larger available reservoir of dissolved silicon compared to interglacial conditions. Yes, the system eventually may run out of dissolved silicon (and thus get into Si-limited conditions, just as you hypothesize), but siliceous producers are at least set to succeed at the start of it all… Boom&Bust at its best… something diatoms are really good at.

Figure 6 and Subsection 5.3.3. in general: the description and interpretation of the data sounds fair and accurate enough, especially when it comes to relative increases/decreases of the various components during glacials and interglacials (i.e., the large-scale features and implications of the records). Nuances in these patterns, and variability between the different sectors, might be however a bit trickier to explain properly as, besides the large overestimation for some fluxes during glacials (thay you nicely presented and argued for, based on the Th corrections), the oceanographic significance of each station is a lot more heterogeneous than simply being representative of "subantarctic conditions" (see Fig. 6a, b). This might not be that drastic for dissolved silicon, as the vast majority of the stations have near zero values, with very few exceptions, but dissolved nitrate is highly variable across the sites, going from just above zero micromole/litre at the top of the SE

Atlantic transect, to almost 30 micromole/litre for the Pacific Sector and the easternmost station in the Indian Sector.

Lines 490-492. The discussion about the productivity response not scaling up to the magnitude of the lithogenic input shift…. I have two related comments to this: it does not necessarily need to scale in the same way… it takes very little additional bioavailable Iron to get out of Fe-limited conditions (two nanomoles or thereabouts?), so any additional lithogenic input, regardless of its source, above and beyond the one required to reach such concentrations in the surface waters, will not change the deplete/replete status of the water column and hence the main characteristics of the productivity system associated with those waters. My other comment is that maybe the main peculiarity of your site (main source of Fe coming from PIS as lithogenic input, not via eolian pathways as elsewhere… and in general the very close proximity to such source and land) plays a big role in this: this would presumably make it easier (and faster) to switch from an Fe-deplete to Fe-replete conditions, thus going very quickly into the "saturated" state I was mentioning above: regardless how much more Fe gets dumped in there, it won't make much of a difference anymore at some point.

**Technical Correction**

Line 32: as well as with a decrease

Lines 48-49: I would change this sentence around to "The Drake Passage (DP), located between the southern tip of South America and the Antarctic Peninsula, is a major constriction for the ACC flow and SO fronts". I would call it "a" constriction, as plateaux and gaps between ridges also exert quite strong disturbances to ACC flow.

Line 51: World Ocean. In the same line, before mentioning its low efficiency, briefly describe what the biologic pump is.

Line 67: Why "the last"? …explaining 30 to 50 ppm of atmospheric CO2 drawdown…

Line 89: their link

Line 100: as a few lines below you are writing about CDW and carbonate undersaturation, it would be useful to mention the site water depth here (3781m, mentioned in line 149).

Line 112: … between silicate-poor waters to the north and silicate-rich waters to the south of it…

Line 119: phosphorus

Line 124-126: Probably perfectly fine as written, but slightly unclear whether the substantial modern dust contribution is also "excluded" or not. Please make it clearer by either switching the two sentences "… have demonstrated a substantial…. and excluded a westward…" or "… have excluded both a westward… and a substantial…".

Lines 131-132: During last glacial, the distance between core PS97/093-2 and the PIS (situated at ~56°S at that time, Glasser et al., 2008) … or: reaching as far south as ~56°S at that time

Legend to Figure 1, last line: remove this "Schlitzer, Reiner, Ocean Data View, odv.awi.de, 2021" as it is the full reference for Schlitzer, 2021.

Line 156: was developed by Toyos et al. (2020) using a two-step approach

Line 164: and homogenized sediment

Line 172: are expressed as biogenic opal percent by

Line 235: Is there something missing in this sentence: "…were calculated by dividing the average 232Th concentration"? Maybe something like: "… by dividing the lithogenic material concentration by the average…"?

Lines 250-255: As you did elsewhere, please mention in the appropriate place the panel/letter, not just Figure 2, as it is sometimes difficult to understand which pattern/curve is being described, especially when you write about "other BMARs".

Legend to Figure 2: TOC and Ba excess have been swapped.

Legend to Figure 4: Would it be useful to specify what the MAR in panel d) refers to in this case? Sediment MAR, Bulk Sediment MAR, or a similar expression.

Lines 325-328: I found these lines difficult to follow… Yes, I do get that they are an explanation of why winnowing, due to cohesive effects, seems to (counter-intuitively) occur during slower current speed intervals, but the period doesn't read properly. Probably this is due to a very long sentence having been split in two, starting with "Since". How about rewording to something like "We suggest this is due to the fact that under strong…., while under slower…."?

Figure 5: Core site PS75/76, and even more so PS75/56, are not really located in Australia/Oceania… Yes, they might receive the far end of dust plumes from Australia but are really in the central South Pacific.

Line 354: "those reconstructions from"… replace with "those observed in", to avoid the more convoluted "those reconstructed for open Pacific… locations/core sites".

Line 356: Thus, in order to account for these very high lithogenic fluxes in the proximity of the DP, an additional non-eolian source of terrigenous material is required.

Line 451: Something is a bit strange in this passage "upwelled Fe south of the PF, because of in the hypothetical case"

Line 459: make the cause/effect relationship clearer/more explicit: "… may be the result of a reduced diatom Si:N…"

Line 466: suggests

Line 467: significantly the integrated

Line 549: leading to calcareous plankton (i.e., coccolithophores) becoming the

Line 551: was perceptible, as demonstrated by the occurrence of a prominent calcareous ooze

Table S1 legend. normalized MARs of export production proxies

---

## Author Comment (AC2)

Dear Editor and Reviewers,

We are grateful for your comments and the constructive feedback and suggestions that helped us to improve the manuscript. We modified the manuscript accordingly, including figures, and supplementary material, and will upload the revised version as per Editor's request. In the attached letter, we outline the response to each of the comments using the following color code:

- Original text from the reviewers in black
- Our response in blue

Because a couple of issues were raised by both reviewers, our extended response can be found under Reviewer 1.

Sincerely yours,

María H. Toyos (on behalf of all co-authors)

**RESPONSE TO REFERRE #1 (G. CORTESE)**

**General Comments**

This manuscript presents new data from a sediment core located close to the Drake Passage. These data allow the authors to discuss lithogenic fluxes to their core site and their influence (as bioavailable Fe) in shaping the character and magnitude of export production in the Subantarctic region of the Southern Ocean, over multiple glacial-interglacial cycles. One innovative component is the fact that all their Mass Accumulation Rates are Th-corrected, which is a great practice as it takes into account focussing/winnowing effects, besides allowing the authors to provide insights on current dynamics, when coupled, as they do, with other methods (i.e., sortable silts). Another important aspect of this study is the comparison to a host of other records available from the different Sectors of the Southern Ocean, allowing the reader to get a circum-antarctic perspective of export production variability through time. Methods are valid and results presented in a very clear, easy to follow style, with appropriate referencing of related work. The paper is nicely written, logically structured and figures and tables are both of good quality and appropriate.

**R**: We appreciate your positive response and thank you for the suggestions made.

**Specific Comments**

Lines 44-46: Fronts as barriers: In principle, yes, but mesoscale dynamics/eddy fields allow a certain level of mixing across fronts.

**R:** We clarified this point by adding a sentence after the original statement:

..."These fronts act as barriers, inhibiting the exchange of the upwelled waters and their associated nutrients with neighboring fronts, and therefore also represent the limits between geochemical provinces (Chapman et al., 2020; Paparazzo, 2016). *Nevertheless, in some regions of the SO with weaker jets, known as "leaky" jet regions, the mixing barrier effect is lower, allowing some degree of meridional exchange of nutrients and upwelled waters by eddy fields (Naveira Garabato et al., 2011).*"...

Lines 159-160: "For tuning, we assumed that low Fe contents characterize interglacial periods, whereas high contents represent glacials". Given how this paper is specifically addressing issues related to Fe-depleted/Fe-replete conditions, it would be good to provide a bit more information on how much (or how little) the tuning distorts the preliminary age model. Or, at least, a more in-depth mention of the supporting evidence for this choice (rationale behind it, and support from other measurements, like carbonate, Ca counts, etc).

**R:** Since there are comments of both reviewers related to the age model, we decided to extend this answer to cover all raised review issues and refer to this answer in the following text:

We thank both reviewers for the comments related to the age model. Our initial age model was originally based on shipboard measurements of magnetic susceptibility and GRA-density. The magnetic susceptibility was, however, measured with a whole-round core logger. This fast method provides reliable but somewhat smoothed records. Therefore, we decided to use the XRF scanner-based high-resolution major element records to get a higher resolution. We chose the Fe record that best represents the total siliciclastic content of the sediment and largely parallels the shipboard magnetic susceptibility record.

Additionally, during the space of time that encompasses this study, biostratigraphic time markers helped us substantially with the age model: Zone NN21 was recognized based on the first occurrence of *Emiliana*

*huxleyi* at 0.29 Ma (between 143 and 43 cm); the last occurrence of *Actinocyclus ingens* together with an ooze of *Gephyrocapsa caribbeanica* at 343 cm indicate the transition from MIS 12 to MIS 11 (see Toyos et al. 2020).

We revised the age model chapter accordingly by clarifying in the text the supporting evidence for this choice (see changes in the paragraph below):

…"The age model for core PS97/093-2 **was developed by**  Toyos et al. (2020) using the AnalysSeries software (Paillard et al., 1996), and it is based in a two-step approach: 1) establishment of a preliminary age model based **on onboard physical property data** and biostratigraphic time markers from calcareous nannofossils and diatoms, and 2) fine-tuning of the high-resolution XRF-derived elemental Fe and Ca counts and CaCO$_3$ contents to the LR04 benthic $\delta^{18}$O stack (Lisiecki and Raymo, 2005)**. As Fe content is generally representative of the sediment's siliciclastic fraction, which is most likely controlled by a combination of factors including dilution with biogenic material (primarily CaCO$_3$), together with varying eolian and/or glaciogenic sediment input from South America.** For tuning, we assumed that low Fe contents characterize interglacial periods, whereas high contents represent glacials. Additionally, XRF Ca counts and CaCO$_3$ percentages were used for additional tuning in the intervals where they are present"…

Lines 176-177: "For terrigenous contents exceeding 75%, opal concentrations measured at UdeC are consistently 3–5% higher than those measured at AWI". Even if the subsequent mention of similar patterns of variability is right on point, it would be useful to mention the potential reasons for such systematic difference.

**(COMMENT OF REFEREE #2 related to the same topic):** I am also intrigued by the consistent difference between % opal measured in the different labs in samples with high terrigenous content. The methods section says sample sizes at UdeC were between 50-70mg. Were they varied intentionally due to changes in terr. content? The alkaline extraction method is sensitive to sample size; for example, for a sample with >75% terr. content, a sample size of 100mg might be warranted. The methods don't say if the same sample sizes were used at AWI. This could account for some of the difference. I am not worried about this in terms of the overall conclusions of the study, but additional details about AWI sample sizes could be included in methods.
**R:** Since there are comments of both reviewers related to this point, we decided to extend this answer to cover all raised review issues and refer to this answer in the following text:

We thank both reviewers for the comments related to the methodological strategy for opal determination. As part of an inter-calibration strategy between UdeC and AWI, Wu et al. (2019), at AWI, re-analyzed the biogenic opal of surface samples from the Drake Passage that were firstly measured at UdeC and published by Cárdenas et al. (2019). Wu et al. (2019) found a good correlation (r$^2$=0.8) between the two datasets and somewhat lower values at AWI. They suggested that the difference between the two laboratories might be related to a different extent of leaching of clay minerals like smectite (Schlüter and Rickert, 1998) and potentially also sample sizes.

With respect to the core used in this study (PS97/093-2), we used 30 mg for opal measurements at AWI whereas at UdeC, we increased the sample size from 50 mg to 70 mg. An additional difference for the disparity between both laboratories is the higher pH base used at UdeC. For this study, we decided to use the opal results from UdeC because of the higher temporal resolution of the opal measurements.

Following the reviewers' advice, we will add the potential reason for such systematic differences in the revised version (see new paragraph below):

…"Biogenic opal was determined at the Laboratory of Paleoceanography, University of Concepción (UdeC), Chile. The alkaline extraction was conducted following the procedure described by Mortlock and Froelich (1989), but using NaOH as a digestion solution (Müller and Schneider, 1993). Between fifty to seventy milligrams of freeze-dried sediment were first treated with 10% $H_2O_2$ and 1N HCl, and then extracted with 1M NaOH (40 mL; pH~13) at 85 °C for five hours. The analysis was carried out by molybdate-blue spectrophotometry. Values are expressed as biogenic opal ***percent*** by multiplying the Si (%) by 2.4 (Mortlock and Froelich, 1989). We did not correct for the release of extractable Si from coexisting clay minerals, and thus biogenic opal values could be overestimated (Schlüter and Rickert, 1998). Biogenic opal was also measured at AWI Bremerhaven using the sequential leaching method of Müller and Schneider (1993) at much lower temporal resolution, and offsets between the overlapping data sets were observed. For terrigenous contents exceeding 70%, opal concentrations measured at UdeC are consistently 3–5% higher than those measured at AWI. When the lithogenic content was below 40%, the inter-lab difference was less than 1%. ***The offset between both datasets is most probably due to leaching clay minerals like smectite at UdeC (Cárdenas et al., 2019; Wu et al., 2019) and a higher pH base employed at UdeC.*** Despite the difference in values, both records show a similar pattern of variability. Given the importance of high-resolution data, we here use the opal results from UdeC."….

Line 208: Does the relatively lower match (compared to Fe and Ca) between XRF and bulk sediment measurements for Ba relate entirely to the two main mentioned uncertainties (Ba as marine barite and assumed constant Ba/Ti ratio for terrigenous components), or are there other factors potentially affecting this deviation?
**R:** Since Ba records from XRF usually are very smooth and without large scatter, we think that the lower correlation, in this case, might generally indicate low Ba contents, and therefore more scatter.

Traditionally, in the XRF-bulk sediment correlations, Fe and Ca in most cases show the best correlation coefficients, and $r^2$=0.7 is generally considered a good value for Barium.

Line 246: Ba excess higher during interglacials: interestingly, mainly true for MIS 5, 7, 9, not so much for either MIS 1 or 11... So that peculiar pattern might have a reason behind it.
**R:** Since $Ba_{xs}$ (wt%) is always higher than in the preceding glacial, instead of writing that "Ba excess higher during interglacials", we re-phrased slightly by saying that it is higher than in the preceding glacial (see new sentence below). As the comment above refers to the $Ba_{xs}$ (wt%) record, which might not be primarily related to paleoproductivity (due to the record is not Th corrected), and the main goal of this paper is to characterize paleoproductivity patterns, we prefer not to discuss in detail patterns in the wt % records.

…"The $Ba_{xs}$ content shows glacial-interglacial variability with higher values during interglacials **than in the preceding glacials**"

Line 250: Carbonate and Ba excess BMAR varying in parallel to their respective percentages. True in general, but different pattern (for both of them) during MIS5e compared to MIS 5a-d. Which, again, might be a peculiarity worth exploring.
**R:** We added a potential reason for this pattern:

…"$CaCO_3$ and $Ba_{xs}$ BMAR records vary in parallel with percentages of carbonate and $Ba_{xs}$**, *except for MIS 5e compared to MIS 5a–d, likely caused by an increase in the sedimentation rates from ca. 0.5 to 2***

*cm/ka during MIS 5e (Figs 2d,g, and 4).* In contrast *with the CaCO$_3$ and Ba$_{xs}$ BMAR records, * all other BMAR of individual components  show no strong similarities with the corresponding percentages *(Fig 2b,c,e,f)."*

Legend to Figure 3: The mention of export production is a bit misleading: these are burial rates in the sediment, as derived by MARs.... By their nature, compared to export production, they integrate the effect of dissolution and remobilization (focussing/winnowing) through the water column and at the water/sediment interface.
**R:** Definition changed.

In the same legend: how the error envelopes were derived should be mentioned somewhere in the text under Methods.
**R:** Ok, done.

Lines 276 and 279: From MIS 7... From mid-MIS 9... To a certain extent, both of these expressions do not really indicate the timing of initiation of a clear pattern but have more to do with the availability of data dense enough as to recognize that pattern in the first place.
**R:** Agreed, clarified.

Line 368: "and strong similarities between the dust peaks in Antarctica and the advances of the PIS". Just as you did for the IRD record, mention explicitly why this observation is relevant to the glaciation status of PIS and its role in increased lithogenic material input to your core site during glacials. You mentioned this in general terms in lines 358-359. This is an important point to clarify also in light of your statements on the EDC dust record being mainly linked to eolian inputs... essentially re-iterating that enhanced glacial conditions in PIS lead to increased terrestrial input to the adjoining ocean. Or you may decide that it is not necessary to repeat this again here, as you describe exactly how this this interplay works when talking about bioavailable iron in lines 378- 380.
**R:** We added a short sentence explaining why this observation is relevant to the glaciation status of PIS and the potential of increased lithogenic material in the Southern Ocean (see new sentence below):

…"strong similarities between the dust peaks in Antarctica and the presence of *active outwash plains in Patagonia*  have been reported by Sudgen et al. (2009) **due to the high sediment load from the glaciers is exposed, and therefore favors the mobilization of the terrigenous material**."

Line 408: "all being lower during full interglacials (Holocene, MIS 5 and MIS 11". Two remarks: You do provide a PAGES statement for this statement, but usually MIS 7 and 9 are considered to be full interglacials as well... Maybe one way to avoid the diatribe of what is a full interglacial and what is not, one could rephrase this to "being lower during some interglacials...". Second remark: the patterns are difficult to argue for MIS 11 as based on just 2 datapoints... While the statement might still be ok for Holocene and MIS 5, it definitely isn't true for Baxs during MIS 11, where the measured value is actually the highest on your record.
**R**: To avoid the diatribe, and taking into account that it is indeed difficult to argue for MIS 11 (due to the low amount of datapoints during this interval), we rephrased the sentence accordingly:

…"all being higher during MIS 6 and MIS 4–2, and lower during  (the Holocene, and MIS 5  (Table S1, Fig 3)."

Lines 412-413: "Since the preservation of organic carbon in sediments is globally scarce (about 1%". Please specify what this 1% refers to: is it the percentage of TOC produced in the water column that gets preserved in sediments, or is it (a sort of globally-averaged) percent content of TOC in sediments? I guess the former, but not sure.
R: Agreed. We specified this in the revised version.

Line 419: "diatom growth". Not 100% sure growth is the right word here... could "diatom frustule multiplication" or "diatom blooms" be better? Also, I think the sentence does not read right... suggestion "... that diatom frustule multiplication was more effective during the LGM compared to today (resulting in higher opal burial during the LGM), as diatoms reduce their Si/C..."
R: Changes made according to the suggestion, however, instead of "diatom frustule multiplication" or "diatom blooms", in the revised version we prefer to say "diatom growth rate", due to diatoms divide vegetatively, and, when resources are not limiting, at a fast rate.

…"Several studies proposed that **diatom growth rate was**   more effective **during the LGM compared to**  **today (resulting in higher opal burial during the LGM), as**  diatoms can reduce their Si/C uptake ratio under Fe-replete conditions (Anderson et al., 2002; Chase et al., 2015)"….

Lines 434-435: In connection to what discussed in my comment to line 408 (your/PAGES statement about "full interglacials", excluding MIS 7 and 9), it would probably be interesting to mention how those two interglacials did not seem to manage to "offset" carbonate dissolution as all other interglacials did in your record...
R: We agree with this comment, it is now mentioned:

"…changes in export production of coccolithophores and foraminifera (Fig. 3). **As a result, in our record carbonates are only present in the globally strong interglacials MIS 11, MIS 5 and the Holocene (PAGES, 2016), and during weaker integlacials, such as MIS 7 and MIS 9, carbonates did not manage to offset carbonate dissolution"**….

Lines 490-492. The discussion about the productivity response not scaling up to the magnitude of the lithogenic input shift.... I have two related comments to this: it does not necessarily need to scale in the same way... it takes very little additional bioavailable Iron to get out of Fe-limited conditions (two nanomoles or thereabouts?), so any additional lithogenic input, regardless of its source, above and beyond the one required to reach such concentrations in the surface waters, will not change the deplete/replete status of the water column and hence the main characteristics of the productivity system associated with those waters. My other comment is that maybe the main peculiarity of your site (main source of Fe coming from PIS as lithogenic input, not via eolian pathways as elsewhere... and in general the very close proximity to such source and land) plays a big role in this: this would presumably make it easier (and faster) to switch from an Fe-deplete to Fe-replete conditions, thus going very quickly into the "saturated" state I was mentioning above: regardless how much more Fe gets dumped in there, it won't make much of a difference anymore at some point.
R: Thanks for the comment, we wrote a sentence adding the points exposed above:

…"Furthermore, the glacial and interglacial export production fluxes are higher in the Atlantic and Indian sectors than at the DP entrance and in the central Pacific (Fig 6). Whereas in the central Pacific, the relatively low export production is explained by weaker Fe fertilization caused by lower local glacial dust fluxes (Lamy et al., 2014), the results at our site indicate that export production did not respond **exclusively**

*to glacial Fe fertilization  (Fig 6).* **Since it takes very little bioavailable iron to get out of the Fe-limited conditions (ca. 3nM, Boyd et al., 2000), and our core site is proximal to Fe sources, which might facilitate a faster transition from a Fe-deplete to Fe-replete conditions.** * We suggest that under bioavailable Fe-replete conditions, another mechanism than Fe fertilization might ultimately regulate export production at our site."*

Lines 512-516: Besides the mechanisms you propose in this section, an additional one could be mentioned (besides frontal, and accompanying nutrient fields, movements and increased Fe supply) as playing a role in pushing/supporting the glacial system towards increased opal productivity during glacials: the better Si/N utilization by diatoms under Fe-replete conditions (that you mentioned elsewhere, with references to Brzezinski&al and Frank&al), which would also provide a larger available reservoir of dissolved silicon compared to interglacial conditions. Yes, the system eventually may run out of dissolved silicon (and thus get into Si-limited conditions, just as you hypothesize), but siliceous producers are at least set to succeed at the start of it all... Boom&Bust at its best... something diatoms are really good at.
**R:** We agree with this comment. Therefore, in the new version, we mention the better S/N utilization by diatoms in this paragraph.

Figure 6 and Subsection 5.3.3. in general: the description and interpretation of the data sounds fair and accurate enough, especially when it comes to relative increases/decreases of the various components during glacials and interglacials (i.e., the large-scale features and implications of the records). Nuances in these patterns, and variability between the different sectors, might be however a bit trickier to explain properly as, besides the large overestimation for some fluxes during glacials (that you nicely presented and argued for, based on the Th corrections), the oceanographic significance of each station is a lot more heterogeneous than simply being representative of "subantarctic conditions" (see Fig. 6a, b). This might not be that drastic for dissolved silicon, as the vast majority of the stations have near zero values, with very few exceptions, but dissolved nitrate is highly variable across the sites, going from just above zero micromole/litre at the top of the SE Atlantic transect, to almost 30 micromole/litre for the Pacific Sector and the easternmost station in the Indian Sector.
**R:** We agree on the heterogeneity of the SAZ, which complicates the interpretation.
However, despite the heterogeneity, in our work, we suggest that nitrate depletion in the SAZ is unlikely because of:
   i.   Previous works in the Atlantic sector of the SAZ (i.e., ODP Site 1090 from Martínez-Garcia et al., 2014) have shown that nitrate levels were never completely consumed during the last glacial, and therefore did not become limiting there.
   ii.  The "relatively" high values of export production happen in the SO's Atlantic sector, which is the area of our map with the lowest nitrate content.

**Technical Correction**

Line 32: as well as with a decrease
**R**: Added.

Lines 48-49: I would change this sentence around to "The Drake Passage (DP), located between the southern tip of South America and the Antarctic Peninsula, is a major constriction for the ACC flow and SO fronts". I would call it "a" constriction, as plateaux and gaps between ridges also exert quite strong disturbances to ACC flow.
**R**: It is true, changed.

Line 51: World Ocean. In the same line, before mentioning its low efficiency, briefly describe what the biologic pump is.
R: Added.

Line 67: Why "the last"? ...explaining 30 to 50 ppm of atmospheric CO2 drawdown...
R: Deleted.

Line 89: their link
R: Changed.

Line 100: as a few lines below you are writing about CDW and carbonate undersaturation, it would be useful to mention the site water depth here (3781m, mentioned in line 149).
R: Added.

Line 112: ... between silicate-poor waters to the north and silicate-rich waters to the south of it...
R: Switched.

Line 119: phosphorus
R: Corrected.

Line 124-126: Probably perfectly fine as written, but slightly unclear whether the substantial modern dust contribution is also "excluded" or not. Please make it clearer by either switching the two sentences "... have demonstrated a substantial.... and excluded a westward..." or "... have excluded both a westward... and a substantial...".
R: Clarified.

Lines 131-132: During last glacial, the distance between core PS97/093-2 and the PIS (situated at ~56°S at that time, Glasser et al., 2008) ... or: reaching as far south as ~56°S at that time
R: Changed by "(situated at ~56°S at that time, Glasser et al., 2008)".

Legend to Figure 1, last line: remove this "Schlitzer, Reiner, Ocean Data View, odv.awi.de, 2021" as it is the full reference for Schlitzer, 2021.
R: True, erased.

Line 156: was developed by Toyos et al. (2020) using a two-step approach
R: Changed.

Line 164: and homogenized sediment
R: Changed.

Line 172: are expressed as biogenic opal percent by
R: Added.

Line 235: Is there something missing in this sentence: "...were calculated by dividing the average 232Th concentration"? Maybe something like: "... by dividing the lithogenic material concentration by the average..."?
R: Added.

Lines 250-255: As you did elsewhere, please mention in the appropriate place the panel/letter, not just Figure 2, as it is sometimes difficult to understand which pattern/curve is being described, especially when you write about "other BMARs".
**R:** We added letters to each panel.

Legend to Figure 2: TOC and Ba excess have been swapped.
**R:** True, changed.

Legend to Figure 4: Would it be useful to specify what the MAR in panel d) refers to in this case? Sediment MAR, Bulk Sediment MAR, or a similar expression.
**R:** Specified.

Lines 325-328: I found these lines difficult to follow... Yes, I do get that they are an explanation of why winnowing, due to cohesive effects, seems to (counter-intuitively) occur during slower current speed intervals, but the period doesn't read properly. Probably this is due to a very long sentence having been split in two, starting with "Since". How about rewording to something like "We suggest this is due to the fact that under strong...., while under slower...."?
**R:** Reworded.

Line 354: "those reconstructions from"... replace with "those observed in", to avoid the more convoluted "those reconstructed for open Pacific... locations/core sites".
**R:** Reworded.

Line 356: Thus, in order to account for these very high lithogenic fluxes in the proximity of the DP, an additional non-eolian source of terrigenous material is required.
**R:** Added.

Line 451: Something is a bit strange in this passage "upwelled Fe south of the PF, because of in the hypothetical case"
**R:** Modified.

Line 459: make the cause/effect relationship clearer/more explicit: "... may be the result of a reduced diatom Si:N..."
**R:** Modified.

Line 466: suggests, and Line 467: significantly the integrated
**R:** Done.

Line 549: leading to calcareous plankton (i.e., coccolithophores) becoming the
**R:** Added.

Line 551: was perceptible, as demonstrated by the occurrence of a prominent calcareous ooze
**R:** Done.

Table S1 legend. normalized MARs of export production proxies
**R:** Deleted.

Figure 5: Core site PS75/76, and even more so PS75/56, are not really located in Australia/Oceania... Yes, they might receive the far end of dust plumes from Australia but are really in the central South Pacific.
**R:** The main reason for adding the continental masses in the figure is to provide an idea to the reader of the sources of lithogenic material that could reach the sediment cores; our goal was not to reflect the distance between sources and core locations in the figures.

To avoid confusion, we added a label indicating the core location in the figure (see the modified figure below):

[Figure]

**RESPONSE TO REFERRE #2 (L. BRADTMILLER)**

The manuscript by Toyos and coauthors presents a new, high-quality, high-resolution record of various biogenic and lithoigenic proxies in the Drake Passage over several glacial cycles. The high resolution of the data and the relatively long (for the Southern Ocean) span of the record are especially useful in this fairly under-studied region. Finally, the fact that their records have been Th-normalized avoids many pitfalls of working in the SO, where sediment focusing and winnowing can be intense. I have relatively brief comments due to 1) the overall high quality of the study and presentation and 2) the incredibly thorough comments already provided by another reviewer.

**R**: We appreciate your positive response and thank you for the suggestions made.

First, I agree with the previous comments that it would be helpful to hear more about how much/little the age model changed as a result of tuning to high/low Fe contents. This could come in the text, or could easily be illustrated with a supplemental figure.

**R**: Please, see the extended response to the comment above.

I am also intrigued by the consistent difference between % opal measured in the different labs in samples with high terrigenous content. The methods section says sample sizes at UdeC were between 50-70mg. Were they varied intentionally due to changes in terr. content? The alkaline extraction method is sensitive to sample size; for example, for a sample with >75% terr. content, a sample size of 100mg might be warranted. The methods don't say if the same sample sizes were used at AWI. This could account for some of the difference. I am not worried about this in terms of the overall conclusions of the study, but additional details about AWI sample sizes could be included in methods.

**R**: Please, see the extended response to the comment above.

Lastly, I appreciate the care taken by the authors to consider the possibility that not all (or, as they point out, a varying amount) of the added Fe was bioavailable. Too many studies in this region assume that any Fe input is an automatic link to productivity. Even if we don't have the tools to reconstruct bioavailble Fe content, including this nuance in the discussion is a welcome change.

In summary, this is a high-quality study and a well-written manuscript. I concur with nearly all of the extensive specific comments by an earlier reviewer, and commend the authors on a very strong contribution.

**R**: Thank you for your positive response and comments.